# Reviews and syntheses: Carbon use efficiency from organisms to ecosystems – Definitions, theories, and empirical evidence

Stefano Manzoni[1,2], Petr Čapek[3], Philipp Porada[4], Martin Thurner[2,5], Mattias Winterdahl[6], Christian Beer[2,5], Volker Brüchert[7], Jan Frouz[8], Anke M. Herrmann[9], Björn D. Lindahl[9], Steve W. Lyon[1,2], Hana Šantrůčková[10], Giulia Vico[11], Danielle Way[12,13]

[1]Department of Physical Geography, Stockholm University, Stockholm, SE-106 91, Sweden
[2]Bolin Centre for Climate Research, Stockholm University, Stockholm, SE-106 91, Sweden
[3]Pacific Northwest National Laboratory, Environmental Molecular Sciences Laboratory, Richland, WA, USA
[4]Plant Ecology and Nature Conservation, University of Potsdam, Potsdam, Germany
[5]Department of Environmental Science and Analytical Chemistry, Stockholm University, Stockholm, SE-106 91, Sweden
[6]Department of Earth Sciences, Uppsala University, Uppsala, Sweden
[7]Department of Geological Sciences, Stockholm University, Stockholm, SE-106 91, Sweden
[8]CUNI Institute for Environmental Studies, Charles University in Prague, Prague, Czech Republic
[9]Department of Soil and Environment, Swedish University of Agricultural Sciences, Uppsala, SE-750 07, Sweden
[10]Department of Ecosystem Biology, University of South Bohemia, České Budějovice, Czech Republic
[11]Department of Crop Production Ecology, Swedish University of Agricultural Sciences, Uppsala, SE-750 07, Sweden
[12]Department of Biology, University of Western Ontario, London, Canada
[13]Nicholas School of the Environment, Duke University, Durham, NC, USA

*Correspondence to*: Stefano Manzoni (stefano.manzoni@natgeo.su.se)

**Abstract.** The cycling of carbon (C) between the Earth surface and the atmosphere is controlled by biological and abiotic processes that regulate C storage in biogeochemical compartments and release to the atmosphere. This partitioning is quantified using various forms of C-use efficiency (CUE) – the ratio of C remaining in a system over C entering that system. Biological CUE is the fraction of C taken up allocated to biosynthesis. In soils and sediments, C storage also depends also on abiotic processes, so the term C-storage efficiency (CSE) can be used. Here we first review and reconcile CUE and CSE definitions proposed for autotrophic and heterotrophic organisms and communities, food webs, whole ecosystems and watersheds, and soils and sediments using a common mathematical framework. Second, we identify general CUE patterns; e.g., the actual CUE increases with improving growth conditions, and apparent CUE decreases with increasing turnover. We then synthesize >5000 CUE estimates showing that CUE decreases with increasing biological and ecological organization – from unicellular to multicellular organisms, and from individuals to ecosystems. We conclude that CUE is an emergent property of coupled biological-abiotic systems, and it should be regarded as a flexible and scale-dependent index of the capacity of a given system to effectively retain C.

## 1 Introduction

Carbon cycling is driven by biological, physical, and chemical processes – vegetation and phytoplankton take up $CO_2$ from the atmosphere and convert it to biomass, decomposers and animals convert organic C to biomass and release it as $CO_2$, and physico-chemical processes redistribute and store C. Many of these processes involve the 'conversion' of C from various sources into biological products and the efficiency of this biological conversion is generally referred to as C-use efficiency (CUE). Low CUE values imply that little C is converted to biomass and

biological products relative to the amount consumed. As a result, less C is retained in the organism and more is released as $CO_2$ or other forms of C, in comparison to circumstances when CUE is high and the organism retains more C. In other words, from this perspective, low CUE is indicative of a more open biological C cycle. Therefore, understanding the degree of variation in CUE – especially along gradients of environmental conditions – is key for quantifying how much C is retained in biomass and potentially in an ecosystem in the long term (Allison et al., 2010; Manzoni et al., 2012; Hessen et al., 2004; Jiao et al., 2014; Sterner and Elser, 2002). However, the connection between CUE and long-term C storage is complex, and mediated by multiple biological, ecological, and physical factors.

For biological systems (organs, individual organisms, or even entire communities), CUE is defined as the ratio between the amount of C allocated to biosynthesis (new biomass and biological products, including e.g., exudates) and the amount of C taken up. While the term CUE was proposed in the mid-1990s in the context of plant C balances (Gifford, 1995), other terms – e.g., 'growth yield' – referring to the efficiency of substrate conversion into biomass had been in use since the early 1900 (Monod, 1949). Now, efficiency definitions are proliferating across many disciplines in biology, ecology, and Earth sciences. While some of these definitions are comparable (and all are deceptively simple), subtle differences often emerge, partly due to conceptual and methodological advances that allow quantification of previously ignored C exchanges. These differences make interpretation of results difficult and complicate cross-disciplinary comparisons.

The main difficulty is to unambiguously define what represents growth, release of extracellular compounds or C storage, and reconcile conceptual definitions with empirical estimates (Geyer et al., 2016; Chapin et al., 2006; Clark et al., 2001). In fact, CUE is a property of the biological system considered under the specific conditions it experiences, and synthetizes various biological processes occurring across a range of spatial and temporal scales in a single variable (Geyer et al., 2016). Because the proportion of growth vs. maintenance respiration, the growth rate, the synthesis and release of products, and the availability of C all vary in time in any organism, CUE is also expected to change. Changes in environmental conditions that favour growth over respiration will shift the balance of C allocation towards biosynthesis (or C storage at the ecosystem level), thus increasing CUE (Manzoni et al., 2017; Öquist et al., 2017; Vicca et al., 2012). Instantaneous responses to an environmental change may also differ from long-term responses involving acclimation and adaptation to the new conditions – both of which can potentially affect C allocation to different metabolic processes and hence CUE (Allison, 2014). In addition to responses to environmental change, metabolic processes also differ across levels of biological organization, leading to decreasing values of CUE as organisms become more complex and require more energetically-expensive structures (DeLong et al., 2010).

While the aforementioned mechanisms can be identified for individual organisms or uniform populations, natural plant, microbial, and animal communities are composed of a number of different organisms whose metabolism may respond differently to environmental drivers. In addition, various interactions among organisms in an ecosystem lead to emergent patterns that are different from the sum of individual contributions. Therefore, by integrating the contribution of individual organisms with a range of different CUE values, patterns in community-level CUE may be different from those expected based on organism-level CUE (del Giorgio and Cole, 1998; Ettema and Wardle, 2002; Geyer et al., 2016). For example, seedlings of conifer species can have a whole-plant CUE around 0.7 (Wang et al., 2015), but conifer forests encompassing a range of tree ages and species exhibit a CUE (defined as net primary productivity/gross primary productivity) of around 0.45 (DeLucia et al.,

2007; Gifford, 2003). Similarly, ecosystem level CUE (defined as net ecosystem productivity/gross primary productivity) emerges from linkages between plants and decomposers and the way both communities process and exchange C (Bradford and Crowther, 2013; Sinsabaugh et al., 2017). Because at the ecosystem level $CO_2$ is released by both autotrophs and heterotrophs, ecosystem CUE values are lower than those of plant communities.

While variability in biological and ecological processes affects CUE at organism-to-ecosystem levels, the efficiency of long-term C storage in ecosystems depends on how much C enters physically protected or chemically recalcitrant compartments or is removed from the system by abiotic transport processes. The more C is removed via, e.g., leaching and lateral transfer (Chapin et al., 2006; Cole et al., 2007), the lower the C-storage efficiency (CSE) of an ecosystem. The term CSE is used here instead of CUE to emphasize that C storage in soils and sediments also depends on abiotic processes that do not 'use' C for their fitness in a manner similar to organisms, or on incomplete C turnover due to hampered heterotrophic activity; e.g., in the uppermost organic layers of forest soils or in peatlands. Moreover, as C is recycled in the soil or sediment system and progressively more C is lost, C accumulation becomes more dependent on physico-chemical protection mechanisms that reduce accessibility of C to decomposers and abiotic removal processes (Schmidt et al., 2011; Canfield, 1994; Mendonça et al., 2017; Stewart et al., 2007).

From these examples (and others that will be presented in the following), it is clear that CUE (or CSE) should be regarded as a flexible quantity that emerges from the underlying biological and abiotic processes at various spatial and temporal scales. Understanding to what degree CUE is stable or variable across scales is important for correct partitioning of C in biogeochemical models, in which these efficiencies are sometimes assumed constant (e.g., microbial CUE), and in other cases the result of modelled C fluxes. Measured CUE and CSE thus offer an opportunity for testing the capacity of models to describe how C is partitioned among different pathways.

With this aim in mind, we synthesize the numerous definitions of C-use and C-storage efficiencies currently employed across levels of biological and ecological organization and spatial-temporal scales, and develop a coherent mathematical framework for these different definitions. Next, we analyse how these efficiencies vary across scales and levels of organization, and how physico-chemical processes that lead to stabilisation or incomplete turnover of organic matter become relevant to evaluate C retention at the whole-ecosystem level. While previous syntheses have investigated drivers of CUE in specific systems (Canfield, 1994; del Giorgio and Cole, 1998; DeLucia et al., 2007; Manzoni et al., 2017; Sinsabaugh et al., 2015; Sterner and Elser, 2002), we focus on scale-dependencies of CUE and CSE across systems, and discuss the limitations that arise in the interpretation of efficiency values due to these scaling issues. Finally, we discuss the relevance of observed trends in relation to our understanding of the C cycle, for informing ecosystem model development, and for overcoming disciplinary boundaries that have led to numerous conceptually similar CUE definitions.

**2 Theory**

**2.1 General carbon balance equations and definitions of C-use and C-storage efficiencies**

In this section, general equations are presented to define C-use and C-storage efficiencies (CUE and CSE, respectively). We use the term CUE for efficiencies that are relevant for biological systems (from individuals to communities), in which C is actually 'used' for functions related to the fitness of organisms, either as individuals

or in communities. In contrast, systems in which both biological and abiotic storage processes occur do not literally 'use' C, but 'store' it and thus the term CSE will be used instead of CUE. The term 'storage' is used instead of 'sequestration' because we do not focus on the long-term stabilization of C, but only on the efficiency of C retention in relation to C inputs. All symbols are defined in Table 1 and all quantities are expressed in carbon units. Table 2 summarizes the different definitions of CUE proposed in the literature, and Table 3 relates C

exchanges rates and fluxes used in the theory section to processes specific to the system under consideration.

For a generic C compartment representing an individual organism or a whole ecosystem with clearly defined boundaries, a general mass balance equation can be written in the form

$$\frac{dC}{dt} = I - O, \tag{1}$$

where $C$ is the amount of C in the compartment, and the balance of inputs $I$ and outputs $O$ determine the rate of change of $C$. Inputs and outputs typically depend on external environmental factors and internal state variables,

and are defined differently for organisms and ecosystems, as discussed in the following. In this general equation, changes in stored C can be linked to the rate of C input. This linkage represents a simple definition for an 'apparent' C-use efficiency ($CUE_A$) – the ratio of C remaining in the system (i.e., $\frac{dC}{dt}$) over C added to the system ($I$). The term $CUE_A$ is used for convenience, noting that $CSE_A$ should be used for systems involving abiotic C exchanges. This is an 'apparent' efficiency because it is calculated solely from C input and change in storage,

without any consideration of the underlying processes that determine the C outputs. As will become clear, this pragmatic definition is insufficient for CUE to have a biological meaning. Based on this definition, Eq. (1) can be recast as

$$\frac{dC}{dt} = CUE_A \times I, \tag{2}$$

where $CUE_A$ describes the fraction of the input that ends up in the organic carbon pool $C$. Expanding the definition of $CUE_A$ using the mass balance Eq. (1), $CUE_A$ can also be defined in terms of input and output rates,

$$CUE_A = \frac{dC/dt}{I} = 1 - \frac{O}{I}. \tag{3}$$

These two equalities allow estimating $CUE_A$ from measured changes in C pool size and C exchange rates. Hence, the apparent CUE is a dynamic quantity that depends on the ratio of output to input rates, or the ratio of change in storage and input rate. For systems in which inputs are larger than outputs, $0 < CUE_A < 1$. In contrast, when outputs are larger than inputs, the system loses mass and $CUE_A < 0$. For biological systems where $C$ represents the biomass of the organism, $CUE_A$ represents the fraction of C uptake contributing to a biomass increase. Simialr

considerations hold for whole ecosystems, and $CSE_A$ is accordingly defined as the fraction of C inputs via photosynthesis and physical transport contributing to C storage in vegetation and soils, or sediments (Alin and Johnson, 2007; Canfield, 1994; Stewart et al., 2007).

**2.2 Carbon balance and efficiency equations for biological systems**

Eq. (3) is not particularly useful to describe how effectively an organism uses C because it does not provide much

mechanistic insight into the processes leading to the allocation of C to storage or output rates. However, Eq. (3) is used to estimate apparent CUE in many practical applications where input, change in storage and/or output are measured – including the estimation of CUE for biological systems. If the observational setup is such that changes

in storage and output rates can be unambiguously attributed to certain processes (e.g., gross growth and respiration), then the apparent CUE estimated from Eq. (3) is also a useful measure of the CUE of that organism. However, in general, a more accurate description of the organism C balance is required to define a biologically meaningful CUE, leading to numerous definitions of CUE (Table 2).

Let us now focus on C compartments representing the biomass of an individual organism or of a community. Here, 'organism' indicates any living entity, ranging from unicellular to multicellular, and including both autotrophs and heterotrophs; regardless of their physiology and size, they are all treated as a C compartment with a well-defined boundary that allows defining inputs and outputs. In Table 3, specific processes for each type of organism are matched to the general C balance terms used below. In this context, the input $I$ represents C uptake or ingestion ($U$), and the output $O$ represents the sum of egestion ($EG$), respiration ($R$), exudation ($EX$), and turnover ($T$) (Sterner and Elser, 2002); i.e., the output rate is expressed as $O = EG + R + EX + T$ (Fig. 1). Distinguishing among these processes is motivated by the different time scales for respiration (seconds to hours) and turnover (minutes to years) processes. Egestion includes C that passes through the guts without being assimilated (faeces); for plants and microorganisms, $EG = 0$. The exudation term may include excretion of C compounds such as extracellular enzymes and polysaccharides, and secondary metabolites in microbial communities (Manzoni et al., 2012; Azam and Malfatti, 2007), dissolved organic carbon (DOC) and mucus in animals and phytoplankton (Darchambeau et al., 2003; Azam and Malfatti, 2007), and rhizodeposits (Hutsch et al., 2002) or C export to symbionts (Hobbie, 2006; Ekblad et al., 2013) in plants. Using these definitions, the C balance Eq. (1) can be re-written in more biologically meaningful terms for an individual organism or community as (Fig. 1),

$$\frac{dC}{dt} = U - EG - R - EX - T = G - T, \tag{4}$$

where $U$ is the uptake rate, $U$-$EG$ is the assimilation rate (i.e., $A$ in Fig. 1), and $G$ is the net growth rate. We now define CUE at the organism level as the ratio between the rate of production of biomass and products ($G+EX$), and the rate of C uptake ($U$),

$$\text{CUE} = \frac{G+EX}{U} = \frac{A-R}{U} = 1 - \frac{EG+R}{U}. \tag{5}$$

As a result, the mass balance equation (4) can be rewritten as,

$$\frac{dC}{dt} = \text{CUE} \times U - EX - T = \text{GGE} \times U - T. \tag{6}$$

With this definition, CUE represents the fraction of C taken up that is allocated to biosynthesis (biomass and products that eventually be exuded), but excluding respired and egested C, which do not contribute to biosynthesis. Including exudates such as enzymes and polymeric compounds in the CUE definition may be motivated by the clear fitness advantage these products have for the organism. Moreover, C storage compounds and osmolytes are also regarded as 'biomass', as they would be measured as cellular material.

Other measures of C conversion efficiency have been proposed (Fig. 1) (Sterner and Elser, 2002): i) assimilation efficiency ($AE = A/U$ = assimilation/uptake), ii) net growth efficiency ($NGE = G/A$ = net growth/assimilation), and iii) gross growth efficiency ($GGE = G/U = AE \times NGE$ = net growth/uptake, see the last equality on the right hand side of Eq. (6)). The GGE can be regarded as a biomass yield or production efficiency, as it considers respired, egested, and exuded C as lost from the organism (Payne, 1970; Manzoni et al., 2012; Campioli et al., 2015), different from CUE, which includes exuded C as a product of the C conversion.

The CUE definition in Eq. (5) is consistent with previous work on plant C budgets (Thornley and Cannell, 2000), but it differs from definitions often used for soil microorganisms where only biomass synthesis is considered and CUE = GGE (Manzoni et al., 2012; Geyer et al., 2016) (Table S3). It is thus important to emphasize that CUE as defined in Eq. (5) is in general higher than GGE, because $CUE = GGE + EX/U$. The difference between GGE and CUE is relevant when $EX$ is large, as in the case of organic C exchanges between roots and plant symbionts (Hobbie, 2006; Ekblad et al., 2013), or in anaerobic metabolism (Šantrůčková et al., 2004). In the oceans, 10-30% of microbial production is released as dissolved organic C, but this figure also includes dissolved C from microbial turnover (Benner and Herndl, 2011; Jiao et al., 2014). For soil microbial communities, the extent of the extracellular enzyme and polysaccharide synthesis is unknown but presumably small compared to the other rates involved, at least in aerobic soils where CUE≈GGE (Frey et al., 2001; Šantrůčková et al., 2004). Therefore, making the distinction between GGE and CUE is less important in these systems (for further discussions in this context, see Geyer et al., 2016).

Respiration in Eq. (5) can be further broken down into growth ($R_{growth}$), maintenance ($R_{maintenance}$), and overflow ($R_{overflow}$) components, the latter including futile cycles and compensation of stoichiometric imbalances that are activated when C cannot be used for growth or maintenance (Cannell and Thornley, 2000; Thornley and Cannell, 2000; van Bodegom, 2007; Russell and Cook, 1995). Hence, CUE can be expressed in terms of physiologically distinct respiration rates as,

$$CUE = 1 - \frac{R_{growth} + R_{maintenance} + R_{overflow} + EG}{U}, \tag{7}$$

which demonstrates that any increase in the maintenance and overflow respiration rates relative to growth respiration due to starvation or environmental stresses decreases CUE (Sect. 4.1).

Finally, combining Eq. (2) and (6) provides the relation between $CUE_A$, CUE, and C losses via exudation and turnover,

$$CUE_A = CUE - \frac{EX + T}{U}. \tag{8}$$

Based on this equation, higher turnover or exudation rate reduce $CUE_A$, but not CUE (Eq. (5)).

**2.3 Carbon balance and efficiency equations for systems including abiotic components**

We argued that CUE can be defined for biological entities that use C to improve their fitness, but that CSE should be defined for systems including abiotic components (or when organic matter turnover is incomplete), for which fitness cannot be defined. Examples of such coupled biotic-abiotic systems are whole ecosystems (terrestrial and aquatic), soils, and sediments, where different biological actors (primary producers, decomposers, herbivores, predators) mediate C cycling in association with abiotic processes such as C transport by advection (Chapin et al. 2006, Cole et al. 2007) and C-mineral interactions (Schmidt et al., 2011; Kaiser and Kalbitz, 2012). For these integrated systems, Eq. (1) should be expanded to include these processes (Fig. 2),

$$\frac{dC}{dt} = U + F_{in} - R - F_{out} = NECB. \tag{9}$$

In Eq. (9), $U$ and $R$ represent respectively the C uptake and respiration rates by the biotic components of the system when considering an entire ecosystem (as in Eq. (4)), whereas $U$ refers to litterfall and C deposition when considering soils and sediments, respectively (Table 3). The $F_{in}$ and $F_{out}$ are C inputs and outputs occurring via

abiotic exchanges of organic and inorganic C in natural ecosystems, but also account for anthropogenic inputs (e.g., manure) and outputs (e.g., harvested products) in managed ecosystems. With reference to ecosystems, the C balance of Eq. (9) can also be expressed in terms of the net ecosystem C balance, NECB (Chapin et al., 2006).

In analogy with Eq. (2) and using the rates defined in Eq. (9), the CSE for the whole system can thus be defined as,

$$\text{CSE} = \frac{\text{NECB}}{U + F_{in}} = 1 - \frac{R + F_{out}}{U + F_{in}}. \tag{10}$$

In a purely abiotic system ($U = R = 0$), Eq. (10) can be simplified to $\text{CSE} = 1 - F_{out}/F_{in}$. In contrast, when the abiotic C rates are negligible ($F_{in} \approx F_{out} \approx 0$), Eq. (10) is simplified to $\text{CSE} = 1 - R/U$ – i.e., the C-use efficiency of the biological components in the system (Eq. (5), with $EG$=0). Based on Eq. (10), CSE>0 when an ecosystem is storing C (e.g., systems with long-term accumulation of C in undecomposed necromass, mineral-associated pools, or sediments). As for Eq. (5), the meaning of the C exchange rates in Eq. (10) depend on the system under consideration – e.g., $U$=GPP for entire ecosystems, but $U$=rate of C input to soils when calculating soil CSE (Table 3).

Substituting the definition of CUE for the biological components into Eq. (10), and assuming dominant biological C losses via respiration, an expression linking the system CSE and the biological CUE is found as,

$$(1 - \text{CSE})(U + F_{in}) = (1 - \text{CUE})U + F_{out}. \tag{11}$$

This relation essentially expresses the C losses from the system in two complementary ways – on the left hand side as the fraction of the total C input that is not stored, and on the right hand side as the fraction of the biotic C input that is not stored plus the abiotic losses.

**2.4 C-use and C-storage efficiencies in relation to empirical data**

Equations (5) and (10) provide general definitions of C-use and C-storage efficiencies, for biological and coupled biotic-abiotic systems, respectively (Table 3). The interpretation of these equations is straightforward when a 'control volume' is clearly identified that allows a meaningful empirical estimation of exchange rates and storage changes at the time scale of interest. For example, the body of an animal allows the identification of rates of ingestion, egestion, respiration, exudation, and net growth that, taken together, close the biomass C balance equation. Even in this conceptually simple case, however, cell turnover is not easily quantified. As such, net growth may be measured, but not gross cell growth – and to actually measure these rates can be challenging. In most cases, defining and separating input and output rates is even more complicated – both conceptually and practically when conducting measurements. For example, closing the C balance of leaves, whole plants or plant communities, and aquatic systems is challenging because both input and output fluxes are in the form of $CO_2$. Net exchange fluxes can be readily measured, but not gross fluxes, complicating the separation of $U$ (gross photosynthetic rate in this case) and $R$ (gross autotrophic respiration rate) – not to mention C exports to other parts of the plant and as exudates. Other challenges arise when separating autotrophic and heterotrophic contributions to a single measured respiratory $CO_2$ flux. Common approaches for measuring C exchange rates relevant for CUE and CSE calculations are presented and discussed in the Supplementary Information and the wide range of spatial-temporal scales involved in illustrated in Fig. 3. In our data collection, we compared systems ranging from individual organism and communities, to soils and sediments, food webs, and whole ecosystems and watersheds (Tables 2 and 3).

**3 Data collection and analysis**

Estimates of CUE for a range of organisms (microorganisms, animals, and individual plants), communities (microbial and plant) and ecosystems have been collected from the literature or calculated based on reported C exchange rates (Table S1). Existing datasets or data collections shown in previous publications are used for CUE of heterotrophic organisms (McNaughton et al., 1989; Manzoni et al., 2017), leaves (Atkin et al., 2015), plant communities (Campioli et al., 2015), whole-terrestrial (Luyssaert et al., 2007) and aquatic ecosystems (Hoellein et al., 2013), and for lacustrine and marine sediments (Alin and Johnson, 2007; Canfield, 1994). New literature data collections are developed for CUE of microbial isolates, individual plants, non-vascular vegetation, food chains, soils, and watersheds. The whole database encompasses 5309 CUE estimates.

To compile the new data collections, we conducted an online search using ISI Web of Science and Google Scholar with keywords including various synonyms of CUE or CSE. We also gathered publications following relevant references in articles and books, aided by the expert knowledge of the authors. Due to the enormous variability in terminology used across disciplines, and the fact that in many cases CUE or CSE were not reported in the papers (but only C exchange rates to calculate them), a systematic search was not feasible. Nevertheless, while not exhaustive, our selection of publications covers a broad range of conditions for each subset of data, enabling detection of general patterns across disciplines and scales.

CUE values are recorded in our database as they were reported in the original publications, and thus reflect variation in environmental conditions (e.g., temperature, water availability) and organism status (e.g., actively growing, energy- or nutrient-limited), as well as methodological confounding factors. To facilitate comparisons across datasets, instantaneous CUE values estimated for leaves and non-vascular plant communities were converted to daily values by assuming an equal duration of day- and night-time, and that respiration rates were the same throughout the whole day. Moreover, plant community and ecosystem C fluxes (Campioli et al., 2015; Luyssaert et al., 2007) were averaged first when estimates from different approaches were reported for a given site and year, and second across years to provide long-term mean fluxes. The large majority of data sets encompass independent data points obtained from different sites or treatments. Some time series are included to illustrate how CUE values change during plant ontogeny or as resources are consumed in soil incubations (these datasets are not included in statistical analyses requiring independent samples). One data set required the conversion of energy-based to C-based fluxes to calculate CUE (McNaughton et al., 1989). Energy flux data [kJ/m$^2$/y] were first converted to dry weights using animal and ecosystem-specific plant-community heat of combustion values (Golley, 1961). Dry weights were finally converted to C-mass units assuming a conversion factor of 0.45 g C/g dry weight.

We aim to illustrate the range of variation in CUE across spatial and temporal scales, and levels of biological and ecological organization, but not to explain the observed variability. This latter goal would require ancillary data on environmental conditions and physiological status that are not available in all studies. Further, a comparison of CUE estimates across these diverse data sources is also challenging because of the contrasting spatial and temporal scales at which measurements were conducted (Fig. 3). As such, and given our aim, we have not attempted to bring individual CUE estimates to a comparable scale. For individual plants and microbial communities, CUE estimation approaches vary, and in some cases GGE was reported. Considering the lack of information on the relevance of exudation rates, for these organisms we grouped published efficiencies under the

label CUE. For plant communities in which biomass increments were measured, we use the term GGE (equivalent to biomass production efficiency, as in Campioli et al., 2015).

While it is not possible to quantitatively and statistically compare CUE estimates across all the scales involved due their different meaning, variations can still be interpreted as a result of scale differences. In subsets of the database in which CUE had been estimated in the original sources at consistent spatial and temporal scales, quantitative comparisons among the median values of each subset are possible, and are conducted using the Kruskal-Wallis test with a significance threshold set at $p$=0.05 (MathWorks, 2011). These subsets are: i) long-term average CUE of plant communities and ecosystems, and CSE of soils (plot-to-field spatial scale and annual-to-decadal scale), and ii) CUE of microbial isolates, soil microbial communities, and aquatic bacterial communities (sample size of a litre or less; time scales in the order of days). Moreover, with the same approach we test differences among the medians of smaller data groups within each subset.

For visual comparison, CUE data are grouped according to the subsystems shown in Fig. 2, and the distribution of the available CUE estimates is shown using box-plots. For each subsystem, some examples are extracted to illustrate specific CUE patterns, and the 90th percentile of each group is calculated to provide an indication of the maximum CUE that a sub-system can achieve.

## 4 C-use and C-storage efficiency patterns

Based on the theory outlined in Sect. 2, we present examples on how measured CUE can be driven by 'true' biological factors that affect C partitioning in organisms, but how apparent CUE can be affected by confounding factors such as biomass turnover rates (Sect. 4.1). We then present a synthesis and discussion of CUE trends along biological and ecological levels of organization (Sect. 4.2), across spatial and temporal scales (Sect. 4.3), and compare systems with and without abiotic transport processes (Sect. 4.4). Finally, we ask to what degree CUE estimates are useful for characterizing C allocation patterns and eventually informing C storage calculations and ecosystem models (Sect. 4.5).

### 4.1 Biological drivers vs. confounding factors of C-use efficiency at the organism and community levels

Various forms of CUE are used to characterize the fate of C inputs into a system. To this purpose, CUE is often estimated by measuring changes in C content of and C inputs to that system (Eq. (3)). If biomass turnover and exudation can be neglected, this 'apparent' CUE is a good approximation of the actual CUE (Eq. (8)), but in most cases biomass turnover is present and hard to quantify – in such a case, $CUE_A$ estimates can be significantly lower than the actual CUE (Hagerty et al., 2014; Grossart and Ploug, 2001). Fig. 4a shows how apparent CUE is expected to decrease with increasing turnover rate in relation to C uptake (Eq. (8)). $CUE_A$ values can in principle become negative when the turnover rate is higher than the growth rate (similar issues arise at the ecosystem and watershed scales, but due to C transport rather than turnover). Fig. 4b illustrates these effects by considering data from two studies on soil systems where turnover rate was manipulated. In the first study (Ladd et al., 1992), the $^{14}C$ glucose initially added to the soil is taken up by microorganisms with a certain actual CUE, but as the incubation progresses, the $^{14}C$ remaining in the microbial biomass decreases partly due to cell turnover. As a result, $CUE_A$ at the beginning of the incubation was higher than after about 100 days. It is also possible that during this period substrates became less available, leading to an increase in maintenance respiration compared to growth respiration

(as discussed in the following). In the second example, biomass turnover was manipulated by controlling soil fauna feeding on soil microorganisms (Frey et al., 2001). When grazers were active, the $CUE_A$ estimated from C accumulation into biomass was lower than in the samples without grazers. However, if CUE was calculated from changes in C substrate (glucose) and respiration, estimates were insensitive to grazing pressure (Frey et al., 2001). Similarly decreasing $CUE_A$ has been found in aquatic bacteria subjected to increasing grazing pressure (Grossart and Ploug, 2001). We therefore expect that for a given experimental setting, higher rates of mortality or predation will lead to underestimation of CUE, when using Eq. (3).

Figure 5 illustrates how the relative importance of maintenance costs (respiration and exudation) as compared to growth respiration alters CUE or GGE. Theoretical predictions are shown in Fig. 5a,b, where two methods often used in models to account for the metabolic costs of maintenance are considered (Thornley and Cannell, 2000). When growth respiration has priority over maintenance respiration, C required to sustain maintenance costs is obtained from the pool of assimilated C. In this case, CUE decreases linearly with increasing maintenance costs and CUE can become negative, because maintenance can cause a net biomass loss (Fig. 5a). As an alternative, C required to fulfil maintenance costs can be directly deducted from the C uptake rate, before C is assimilated and available for growth respiration. In this case, CUE can at the lowest reach zero, when all the C taken up is used for maintenance (Fig. 5b). Thus, both modelling approaches yield the same result that CUE decreases with increasing maintenance costs. Empirical evidence lends support to the prediction that maintenance costs decrease the overall CUE, whereas actively growing organisms in which growth respiration is dominant have higher CUE (Sinsabaugh et al., 2015; del Giorgio and Cole, 1998). This simplified view explains some, but not all observed patterns in CUE. For example, low-resource environments could select for high-CUE organisms despite low growth rates. At the other end of the resource availability spectrum, to achieve high growth rates, it might be necessary to increase respiratory losses via C-overflow, futile cycles, and increasing costs of protein turnover, or due to the low energy content of the substrate being consumed. Thus, at very high $G$, a trade-off between growth and CUE may emerge (Lipson, 2015). Combining these pieces of evidence, CUE would be expected to first increase with increasing $G$, then reach a peak and decrease at high $G$ values.

The effect of increasing exudation rate on CUE varies depending on how such increases are realized. If the increase in $EX$ is fuelled by a correspondingly higher $U$, CUE also increases; however, if the increase in $EX$ occurs at the expenses of $G$, such that $G+EX$ is constant for a given $U$, CUE will not be affected. In both scenarios, higher $EX$ decreases the net biomass production, and hence lowers GGE. For example, consistent with these expectations, the microbial CUE values of an aerobic soil (where exudation was negligible) and an anaerobic soil (where exudation was ≈2/3 of the net biomass increment), were comparable (respectively 0.73 vs. 0.70), because the sum of exudation and biomass production were similar (Šantrůčková et al., 2004). However, the GGE of the aerobic soil was much higher than in the anaerobic soil (0.72 vs. 0.43).

For example, the CUE and GGE of microbial communities tend to be low just after addition of a labile C substrate (a lag phase, which can vary in length depending on the preceding physiological status), then to increase sharply as growth rate increases, and finally to decrease because microbes switch from a relatively efficient growth mode when substrates are available to a less efficient maintenance mode when substrates have been exhausted (Öquist et al., 2017) (Fig. 5c,d). Notably, when reductions in biomass occur under starvation conditions due to catabolic conversion of biomass to cover maintenance respiration, CUE<0 (Fig. 5d). Similarly, crops maintain a high CUE until they stop growing vegetative tissues, which senescence while resources are

translocated to seeds (Fig. 5e). In forests, GGE (defined as biomass accumulation over GPP; see Table 2) declines with decreasing nutrient availability (Fig. 5f). However, different from other examples in Fig. 5, this decline

cannot be attributed to higher respiration under nutrient limited conditions, but more likely to higher C investment in plant symbionts (Campioli et al., 2015; Baskaran et al., 2017). Because the effects of higher maintenance respiration or exudation rate have the same direction – both decreasing GGE – we can expect that along resource or environmental gradients characterized by increasing maintenance costs (including exudation), GGE will decrease. Along the same gradients, CUE would decrease only if maintenance costs increase, while it would be

unaffected by changes in exudation rates alone.

A somewhat similar argument has been proposed to explain increases in GGE with increasing nutrient-to-C ratios of the resources used by heterotrophic organisms (Manzoni et al., 2017, where the term CUE was used under the implicit assumption CUE≈GGE). High nutrient availability with respect to C allows growth of the nutrient-rich cells typical of heterotrophs in C limited conditions. However, under nutrient shortage and assuming

that cell nutrient concentrations are relatively stable (homeostasis), resources contain C in excess, which can be selectively removed via overflow mechanisms (Russell and Cook, 1995; Boberg et al., 2008), increased C excretion (Anderson et al., 2005), and possibly C investment in extracellular compounds that promote resource availability (Middelboe and Sondergaard, 1993). As a result, C losses can become decoupled from growth, leading to reduced GGE under conditions of nutrient shortage (Manzoni et al., 2017).

**4.2 C-use efficiency across levels of biological and ecological organization**

We start from the C balance of leaves and move towards whole organisms, communities, food webs, and entire ecosystems to illustrate how CUE varies across levels of biological and ecological organization. The majority of C taken up by leaves is converted into products (CUE≈0.8, Fig. 6a), as might be expected for the organ responsible for entry of C into the biosphere. While leaves only have to support their own limited metabolic needs, whole

plants require energy to maintain a range of additional functions, including nutrient uptake and use, regulation of ion balances and phloem transport (Cannell and Thornley, 2000; Thornley and Cannell, 2000). Thus, the cost of maintaining a complex organism reduces CUE from leaf-level values around 0.9 to whole-plant values of 0.6 (maximum CUE≈0.85). Similarly, moving from unicellular to multicellular organisms requires additional C costs to maintain the structures of increasingly complex bodies (DeLong et al., 2010), resulting in a declining average

CUE from approximately 0.5 (maximum CUE≈0.7) to 0.15 (maximum CUE≈0.5, Fig. 6b,e).

Comparing terrestrial bacteria and fungi, it has been suggested that they should differ in CUE, mostly due to their contrasting life histories (fast growing, inefficient bacteria vs. slow-growing, efficient fungi). Although this paradigm has been around for some time, the hypothesis was not unequivocally supported (Thiet et al., 2006; Six et al., 2006). Recently, fungi and Gram-negative bacteria have been suggested as important

biomarkers when evaluating CUE (Bölscher et al., 2016), but the link between the two is so far not clearly established. The collected CUE data for litter decomposers (arguably mostly fungi, at least in the first phases of litter degradation) suggest a lower CUE than in bacterial communities (Fig. 6b). However, litter decomposers in forest ecosystems face strong stoichiometric imbalances and CUE estimates for these organisms represent long-term averages including periods of slow growth (Manzoni et al., 2017). These factors could explain the lower

average CUE of litter decomposers and aquatic microorganisms compared to soil microorganisms and bacterial isolates – these patterns are thus driven by environmental effects, in addition to organism complexity *per se*.

Food webs include interacting organisms that exchange C among them and with the environment. Each organism exchanges C according to its own CUE (for a modelling example, see Frouz et al., 2013b), but also provides C to the next organism (consumer or predator) in the food web. The C transfer efficiency, defined as the growth rate of a target organism over the rate of C entering the food web (Sect. 1.5 in the Supplementary Information), is then expected to be lower than the CUE of the constituent organisms, as C is lost at each step in the food web (Fig. 6e). Moreover, antagonistic interactions in a food web may increase metabolic costs, also lowering CUE (Toljander et al., 2006). Similar to the organism-level responses to resource availability, also C transfer efficiencies tend to be higher in resource-rich environments. For example, the fish-to-phytoplankton production ratio is higher under nutrient-rich conditions (Dickman et al., 2008). In soils, a food web developing on low C:N litter can be more efficient at retaining C in the system than one developing on high C:N litter, despite no observable difference in C input (Frouz et al., 2013a). In the latter example, it is important to emphasize the role of soil fauna in mediating this response to nutrient availability – the presence of macro-fauna facilitates the transport of C towards sites where it can be stabilized (via bioturbation).

Like moving from simple towards more complex organisms or from single individuals to interacting organisms in food webs, consideration of whole ecosystems also results in lower C retention capacity compared to individual organisms and communities. Aggregation of processes results in a lower CUE for a given GPP, in particular when adding more heterotrophic components (Fig. 2). In fact, including the contribution of heterotrophic respiration is expected to decrease ecosystem CUE compared to vegetation CUE because more of the C taken up by plants is returned to the atmosphere (Fig. 7; Eq. (4) in the Supplementary Information). The CUE of vascular plant communities is indeed significantly higher (CUE $\approx 0.4$) than that for ecosystems (CUE$\approx 0.2$). Ecosystem CUE is expected to be lower than the soil C storage efficiency, because ecosystem CUE is the product of soil CSE (=NECB/NPP, with NPP: net primary productivity) and vegetation CUE (=NPP/GPP, with GPP: gross primary productivity; Supplementary Materials Section 1.6). This is not the case in the data presented in Fig. 7, where the median soil CSE is significantly lower than the median ecosystem CUE. This unexpected result could be explained by the fact that in agricultural systems such as those we used to estimate soil CSE, soil disturbance strongly reduces C accumulation compared to a natural system.

We can also ask how the CUE of individual ecosystem components affects the overall ecosystem CUE. It could be argued that with more efficient organisms, the ecosystem-level CUE would increase, resulting in larger C accumulation (for soil systems, see Cotrufo et al., 2013). There is indeed evidence that microbial communities with higher CUE enhance soil C storage in terrestrial systems (Kallenbach et al., 2016). However, decomposers alter the kinetics of decomposition via extracellular enzymes that are thought to be produced in proportion to the living biomass (Schimel and Weintraub, 2003). As a result of these feedbacks, it is possible that lower (rather than higher) decomposer CUE increases ecosystem CUE and thus C storage potential, as indicated by empirical studies in boreal forests (Kyaschenko et al., 2017) and modelling results (Allison et al., 2010; Baskaran et al., 2017).

Comparing aquatic and terrestrial systems, ecosystem CUE and soil or sediment CSE exhibit contrasting patterns. While the CUE of aquatic ecosystems is significantly lower than that of terrestrial ecosystems (Fig. 6c), the CSE of lacustrine and marine sediments is significantly higher than that of soils (Fig. 6d). The first pattern is explained by allochthonous C contributions to respiration (Sect. 4.3). This explanation should hold also considering that aquatic ecosystem CUE are calculated from daily fluxes, whereas terrestrial ecosystem CUE are calculated from long-term (inter-annual) mean fluxes. In contrast, the higher CSE of sediments than of soils can

be explained by the often high sedimentation rate (Calvert et al., 1992) and the relatively short exposure time to oxygen after burial of organic C (Canfield, 1994; Hedges et al., 1999), whereas most soils remain aerobic and C storage capacity may saturate (Stewart et al., 2007). Indeed, paddy soils where respiration is low due to anaerobic

conditions store C more efficiently (median CSE=0.07) than other agricultural soils (median CSE=0.02; $p$<0.05). Moreover, physical losses from soils (leaching, erosion) are probably larger than for sediments, at least in stable depositional environments.

Based on these analyses we can conclude that higher levels of biological or ecological organization generally imply a more open C cycle – this is caused by increasingly costly structures to maintain complex

organisms, and by increasing heterotrophic contributions when assessing the C storage potential of ecosystems as opposed to primary producers alone.

**4.3 C-use efficiency across spatial and temporal scales**

Moving up spatial and temporal scales involves integrating C exchange rates in space and time. In turn, integrating these exchange rates essentially averages out the contributions at the smaller or shorter scales by considering a

larger number of organisms (e.g., populations vs. individuals) or species (communities vs. populations), a larger spatial domain, and longer periods of time. This averaging effect generally leads to lower CUE than at the smaller scales. As shown in Fig. 3, CUE is estimated over a range of spatial and temporal scales depending on the system of interest, which requires us to interpret CUE in the light of averaged C exchange rates at these scales.

Because organism-level CUE estimates are biased towards actively growing individuals often isolated in

highly controlled conditions, spatial averaging under field conditions, where also inactive or slowly growing individuals are included results in lower population- or community-level CUE. In the case of plants, CUE of individuals is on average around 0.6, whereas in plant communities GGE≈0.4 (Fig. 6a). Part of this difference might be attributed to exudation rates that cause CUE>GGE, but other interpretations are also possible. Quoting Gifford (2003), "The difference may be an expression mostly of the impact of recurring stresses and resource

limitations and the much greater average age of plants in the forests than in the controlled environments. Presumably the respiratory requirement for acquiring water and nutrients is lower when they are abundantly available." (p. 179-180). Moreover, antagonistic interactions within communities might increase C costs (Toljander et al., 2006). This contrast between CUE estimates at individual and community scales is not apparent when comparing CUE of microbial isolates and soil microbial communities, which are not statistically different

(CUE≈0.45, Fig. 6b). However, CUE of aquatic microbial communities from our dataset is significantly lower than that of microbial isolates (CUE≈0.25), despite the occurrence of high values in some communities (del Giorgio and Cole, 1998). The high CUE of soil microbial communities could be due to generally higher resource availability in soils than in aquatic environments, or to amendment of soils with labile compounds that stimulate microbial activity and mask the contribution of slow-growing organisms (Sinsabaugh et al., 2013).

Integrating C exchange rates through time also tends to lower CUE with respect to short-term measurements often conducted after adding labile substrates to heterotrophic systems (Fig. 5c), or during active growing periods for plants (Fig. 5e). Instead, long-term CUE (assuming biomass turnover is correctly accounted for) includes periods of slow growth due to unsuitable environmental conditions, during which maintenance costs remain high while growth stagnates. As mentioned in Sect. 4.2, this could explain why long-term CUE of litter

microorganisms is lower than microbial CUE measured over short periods in other systems (Fig. 6b).

**4.4 Interpreting C-use and C-storage efficiencies in systems with abiotic and anthropogenic C fluxes**

Transport processes can decouple local GPP from ecosystem respiration by feeding heterotrophs with allochthonous C or removing products of primary productivity before they enter the decomposition and herbivory pathways. Allochthonous C can cause relatively large respiration losses even with low inputs from GPP (Duarte and Prairie, 2005; Hoellein et al., 2013), resulting in low or negative values of CUE when defined as NEP/GPP. A more useful definition of CUE should account for allochthonous C inputs, which are however seldom measured (Eq. (10)). This pattern is apparent when comparing the CUE of terrestrial and inland aquatic ecosystems (Fig. 6c) – the former being predominantly positive, the latter being most often strongly negative. Despite inland aquatic systems having negative ecosystem CUE due to large allochthonous inputs, marine systems can act as C sinks due to long-term storage in sediments (where C storage in the range 0.01-0.4% of net primary productivity; Seiter et al. (2005), Falkowski (2014)), as well as accumulation of dissolved inorganic C.

Physical removal of C also alters the estimated CSE. Because physical removal reduces the C that can be stored for a given uptake rate, CSE decreases with increasing abiotic losses of C ($F_{out}$). When these losses of C are considered in addition to respiration, CSE decreases with respect to the ecosystem CUE estimated from biological fluxes, as shown at the ecosystem- and watershed-scale respectively by Eq. (5) and (6) in the Supplementary Information. Using the few available watershed-scale studies where C losses via leaching and subsequent advection in surface water bodies were measured, we can compare CSE estimates with and without the contribution of abiotic lateral C losses. When only the biological components are considered, we found an average ecosystem CUE=0.137, whereas including abiotic losses leads to CSE=0.104 – i.e., a >30% reduction in efficiency. Similarly, in marine systems the export of particulate C from the euphotic zone by particle sinking lowers the potential efficiency of C storage in that zone, while allowing long term storage in the sediments (Dunne et al., 2005).

A large fraction of land and of marine systems is managed to extract food and fibre to support a growing human population (Krausmann et al., 2013). Management of ecosystems has two contrasting effects on CSE, depending on the balance of harvest removal, improved production, and organic amendments. On the one hand, extracting harvested products ($F_{out} > 0$) lowers CSE because a lower fraction of GPP remains in the system. For example, assuming a crop harvest index ranging from 25 to 50% of aboveground biomass (e.g., Unkovich et al., 2010) and a 30% allocation to roots, the percentage of NPP harvested and the corresponding reductions in CSE would range from 17 to 33% (Eq. (10)). On the other hand, management may improve CSE by increasing the production efficiency of vegetation (Campioli et al., 2015), or involve addition of organic C to fields ($F_{in} > 0$; e.g., manure or biochar). These C amendments increase CSE for given respiration and harvest rates, not only thanks to their direct effect through $F_{in}$, but also thanks to indirect effects when soil amendments promote plant productivity. However, this positive effect lessens as the amended organic C is respired and soil organic C reaches saturation levels (Stewart et al., 2007).

**4.5 Do we need C-use efficiency estimates?**

The practical difficulties in estimating CUE at various scales, and the inherent conceptual challenges with its multiple definitions beg the question as to whether it is useful to even discuss CUE. On one hand, there are theoretical and conceptual advantages for using CUE as a 'macroscopic' parameter characterizing organism or ecosystem metabolism – even without quantifying the underlying drivers (specific metabolic pathways, or detailed

input and output rates). In fact, by focusing on the conversion of C into new products rather than on C fluxes *per se*, CUE and CSE patterns offer alternative insights on the inner workings of the processes regulating the C cycle. On the other hand, full process understanding requires identification of these drivers and in such a case, CUE is merely the result of their combination, and knowledge of CUE values alone would be of little use.

        The CUE is less variable than the rates of C exchange that define it and therefore allows comparing
systems characterized by very different C exchange rates. For example, respiration and growth rates of microbial communities roughly double every 10 °C increment in temperature, whereas CUE changes much less – ranging from a 25% decrease for every 10 °C temperature increment (Frey et al., 2013) to no change at all (Dijkstra et al., 2011), depending on the CUE estimation method. Relatively stable efficiencies are particularly useful for modelling purposes, as they allow 'closing' otherwise open (i.e., undetermined) mass balance equations.
Similarly, while NPP, GPP, and ecosystem respiration vary by two orders of magnitude across biomes (Fernandez-Martinez et al., 2014), CUE values are relatively more constrained (if we exclude ecosystems with negative NEP).

        Moreover, non-dimensional numbers – such as CUE and CSE – emerge as key drivers of system dynamics (Vogel, 1998; Buckingham, 1914; Feng et al.). For example, CUE appears in stoichiometric equations describing nutrient fluxes in relation to C fluxes (Manzoni et al., 2010; Sterner and Elser, 2002). In these
stoichiometric models, it is often not necessary to distinguish among various respiration components or to define specific kinetic laws for C exchange rates – a single 'macroscopic', lumped CUE parameter is sufficient to describe the balance of growth and respiration. However, if CUE varies through time or in response to environmental conditions in complex ways, the advantages of having a single lumped parameter may be overcome by a cumbersome parameterization to describe these effects.

A similar issue arises when implementing biological processes that could result in variable CUE into models of soil biogeochemical processes (Allison et al., 2010; Frey et al., 2013), the marine C cycle (Dunne et al., 2005), or vegetation dynamics (Huntingford et al., 2017; Smith et al., 2016). These models differ widely in the way they parameterize the C cycle. For some components of the ecosystem, certain models assume constant CUE values (e.g., CUE of microbial decomposers), whereas for others, more detailed descriptions are employed,
resulting in flexible CUE (e.g., separating respiration components in vegetation) (Gifford, 2003). Empirically established patterns of variation in CUE thus help identification of systems and conditions under which CUE is indeed stable or, in contrast, when additional processes driving variable CUE must be accounted for in models. For example, if soil biogeochemical models are parameterized using microbial CUE values obtained from laboratory incubations, erroneous predictions could be made if those incubations are not representative of soils
under natural conditions. Apart from possible scale mismatches between empirical estimates of CUE and model interpretation, models that assume a stationary set of metabolic responses could underestimate C retention. This is the case when CUE acclimates and buffers the consequences of climatic changes by reducing C losses from the biosphere. In contrast, if changes in CUE amplify biosphere responses – e.g., due to selection of inefficient early-successional species – these models might underestimate potential feedbacks between the biosphere and global
climate.

        In addition to the correct attribution of changes in CUE to processes or environmental conditions, it remains critical to match the definition of CUE used by empiricists with that implemented in models. Specifically, are the same biosynthesis components (e.g., biomass increment vs. exudate export) accounted for in both empirical efficiency estimates and in the model equations? Are abiotic C exchanges at the ecosystem scale both included in

empirical estimates of CSE and described by models? As CUE and CSE represent emerging properties of organisms and ecosystems, they are appealing for model testing, but without a consistent definition, comparisons of model outputs and empirical estimates are not meaningful.

**5 Conclusions**

We have synthesized definitions of and explored variations in the efficiency of C use by organisms, communities
and ecosystems, and in the efficiency of C storage in soils and sediments. This synthesis highlighted conceptual similarities in the way these efficiencies are defined across disciplines, and some common terminological and interpretation issues. In particular, the same term CUE (but also other synonyms) is often used at organism-to-community scales to indicate actual C-use efficiency (Eq. (5)), apparent C-use efficiency (related but not equal to CUE, Eq. (8)), and gross growth efficiency. This mixed use may cause misinterpretations, as it is not clear whether
turnover and biological products are included in the CUE calculations. Similarly, at the ecosystem scale the term CUE is used without specifying whether abiotic and anthropogenic fluxes are accounted for. For improved clarity, we suggest to always define how CUE is estimated with particular attention to C exchanges other than biomass increments and respiration.

Our synthesis shows that turnover deflates 'apparent' CUE estimates, but not 'actual' CUE calculated as
biosynthesis over C uptake ratio. Improving growing conditions generally increases CUE and CSE because it promotes growth processes over C loss processes. Finally, CUE tends to decrease with the level of ecological organization – e.g., from rapidly growing individual organisms to natural communities and ecosystems – as less efficient individuals are considered in communities and more heterotrophic components are sequentially added to the system. Because CUE and CSE are outcomes of a wide spectrum of processes, they are expected to be flexible
and to respond to both biological (e.g., trends in growth vs. respiration) and physical controls (e.g., C transport and environmental conditions). As such – and provided that empirical and model definitions of these efficiencies are consistent – they are useful indices of changes in the C cycle through time and space and could be employed to benchmark short- (in the case of CUE) and long-term predictions (CSE) of soil and ecosystem models.

**Data availability**

The datasets supporting the results that are not already published will be archived in the open-access database of the Bolin Centre for Climate Research (https://bolin.su.se/data/Manzoni-2018).

**Author contribution**

This work was designed by all the authors during two workshops. SM led the writing of the manuscript and prepared the figures; SM, PČ, PP, MT, MW, and GV collated and analysed data from the literature; all authors
contributed to locating literature sources, discussing the content, and writing the manuscript.

**Competing interests**

The authors declare that they have no conflict of interest.

**Acknowledgements**

Funding was provided by the Bolin Centre for Climate Research (Research Area 4), through the project "Scaling carbon-use efficiency from the organism- to the global-scale", and by the Swedish Research Councils, Formas (grant 2015-468 to BL and SM) and Vetenskapsrådet (grants 2016-04146 to BL and SM, 2016-06313 to SM, 621-2014-4266 to CB, and 2016-04910 to GV). HS acknowledges MEYS CZ grants LM2015075 and EF16_013/0001782 – SoWa Ecosystems Research, and DAW acknowledges support from an NSERC Discovery Grant; GV acknowledges the project "TC4F – Trees and Crops for the Future". We also thank all site investigators, their funding agencies, the various regional flux networks and the FLUXNET project, for support to the development of the database "Global Forest Ecosystem Structure and Function Data For Carbon Balance Research" (Luyssaert et al., 2009).

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

**Table 1. Definition of symbols and acronyms.**

| Symbols and acronyms | Description | Dimensions * |
|---|---|---|
| AE | Assimilation efficiency | - |
| BPE | Biomass production efficiency | - |
| $C$ | Carbon-mass | $M\,L^{-2}$ or $M$ |
| CSE | Carbon-storage efficiency | - |
| CUE | Carbon-use efficiency | - |
| $CUE_A$ | Apparent carbon-use efficiency | - |
| $EG$ | Egestion | $M\,L^{-2}\,T^{-1}$ or $M\,T^{-1}$ |
| $EX$ | Exudation | $M\,L^{-2}\,T^{-1}$ or $M\,T^{-1}$ |
| $F_{in}$ | Abiotic carbon input | $M\,L^{-2}\,T^{-1}$ |
| $F_{out}$ | Abiotic carbon output | $M\,L^{-2}\,T^{-1}$ |
| $G$ | Growth | $M\,L^{-2}\,T^{-1}$ or $M\,T^{-1}$ |
| GGE | Gross growth efficiency | - |
| GPP | Gross primary productivity | $M\,L^{-2}\,T^{-1}$ |
| $I$ | Input | $M\,L^{-2}\,T^{-1}$ or $M\,T^{-1}$ |
| NECB | Net ecosystem carbon balance ($= dC/dt$) | $M\,L^{-2}\,T^{-1}$ |
| NEP | Net ecosystem productivity | $M\,L^{-2}\,T^{-1}$ |
| NGE | Net growth efficiency | - |
| NPP | Net primary productivity | $M\,L^{-2}\,T^{-1}$ |
| $O$ | Output | $M\,L^{-2}\,T^{-1}$ or $M\,T^{-1}$ |
| $R$ | Respiration | $M\,L^{-2}\,T^{-1}$ or $M\,T^{-1}$ |
| $R_a$ | Autotrophic respiration | $M\,L^{-2}\,T^{-1}$ or $M\,T^{-1}$ |
| $R_{growth}$ | Growth respiration | $M\,L^{-2}\,T^{-1}$ or $M\,T^{-1}$ |
| $R_h$ | Heterotrophic respiration | $M\,L^{-2}\,T^{-1}$ or $M\,T^{-1}$ |
| $R_{maintenance}$ | Maintenance respiration | $M\,L^{-2}\,T^{-1}$ or $M\,T^{-1}$ |
| $R_{overflow}$ | Overflow respiration | $M\,L^{-2}\,T^{-1}$ or $M\,T^{-1}$ |
| $T$ | Biomass turnover | $M\,L^{-2}\,T^{-1}$ or $M\,T^{-1}$ |
| $U$ | Carbon uptake | $M\,L^{-2}\,T^{-1}$ or $M\,T^{-1}$ |
| $Y_G$ | Growth yield | - |

* M: mass, L: length, T: time, -: non-dimensional quantity.


**Table 2. Summary of definitions of carbon-use and carbon-storage efficiencies. We are not aware of specific terms used with reference to some of the definitions, as indicated by NA.**

| Level of organization | System | Rates/fluxes involved * | Term | Definition | Sources |
|---|---|---|---|---|---|
| Organ | Leaf | GPP, NPP, $R_a$ | NA | $\dfrac{\text{Net photosynthesis}}{\text{Gross photosynthesis}}$ | This paper |
| Organism | Heterotrophic microorganisms | $U, G, R_h$ | Yield, apparent yield, growth yield, C use efficiency, growth efficiency | $\dfrac{\text{Biomass production}}{\text{C uptake}}$ | (Payne 1970, van Bodegom 2007)(Monod, 1949) |
| | Animals | $U, G, R_h$ | Gross growth efficiency | $\dfrac{\text{Biomass production}}{\text{C ingestion}}$ | (Sterner and Elser 2002, Doi et al. 2010) |
| | Plants | $U, G, R_a$ | C use efficiency | $\dfrac{\text{Biomass production}}{\text{Gross photosynthesis}}$ | (Gifford, 1995) |
| Community | Terrestrial microorganisms | $U, G, R_h$ | C use efficiency | $\dfrac{\text{Biomass production}}{\text{C uptake}}$ | (Manzoni et al. 2012, Geyer et al. 2016) |
| | | $U, G, R_h, EX$ | Substrate use efficiency | $\dfrac{\text{Biom. +exudate prod.}}{\text{C uptake}}$ | (Schimel and Weintraub, 2003) |
| | Aquatic bacteria | $U, G, R_h$ | (Gross) growth efficiency | $\dfrac{\text{Biomass production}}{\text{C uptake}}$ | (del Giorgio and Cole 1998) |
| | Plants | GPP, NPP, $R_a$ | Biomass production efficiency | $\dfrac{\text{NPP}}{\text{GPP}}$ | (Cannell and Thornley, 2000; DeLucia et al., 2007) |
| Ecosystem | Soil | NECB, NPP | C sequestration efficiency | $\dfrac{\text{C accumulation rate}}{\text{C input rate}}$ | (Stewart et al. 2007) |
| | Sediments | NECB, rate of C burial | Organic C burial (or preservation) efficiency | $\dfrac{\text{C accumulation rate}}{\text{C input rate}}$ | (Canfield 1994, Alin and Johnson 2007) |
| | Vegetation and soil | NEP, GPP | C use efficiency | $\dfrac{\text{NEP}}{\text{GPP}}$ | (Fernandez-Martinez et al. 2014) |
| | Oceanic photic zone | NPP, rate of C export | Particle export ratio | $\dfrac{\text{C export}}{\text{NPP}}$ | (Ducklow et al., 2001; Dunne et al., 2005) |
| | Food webs (producers, consumers, predators) | $U, G, R_h$ | C transfer efficiency, food chain efficiency | $\dfrac{\text{Biomass production}}{\text{GPP}}$ | (Lindeman 1942, Sterner and Elser 2002) |

| | | | | |
|---|---|---|---|---|
| Watersheds (vegetation, soil, water bodies) | NECB, GPP | NA | $\dfrac{NECB}{GPP + F_{in}}$ | This paper |

* Symbols and acronyms refer to fluxes depicted in Fig. 1 and 2.

**Table 3. Processes associated to the terms of Eq. (5) and (10) at different levels or organization (indicated as subscripts).**

| System | Inputs | | | Outputs | | | |
|---|---|---|---|---|---|---|---|
| | $U$ | $G$ | $F_{in}$ | $R$ | $T$ | $EX$ (and $EG$) | $F_{out}$ |
| Leaves $CUE_{leaf}$ | Gross photosynthesis | Net photosynthesis | - | Dark respiration, photo-respiration | Senescence | | - |
| Micro-organisms $CUE_{microbial}$ | Organic C uptake | Net biomass production | - | Growth, maintenance, overflow respiration | Cell decay, predation | Extracellular poly-saccharides and enzymes | - |
| Animals $CUE_{animal}$ | Food ingestion | Net biomass production | - | Growth, maintenance, overflow respiration | Mortality, predation | Mucus and DOC exudation (and egestion) | - |
| Plants $CUE_{plant}$ | Gross photosynthesis | Net primary productivity | | Growth, maintenance | Mortality, senescence, herbivory | Root exudates, C export to symbionts | Loss due to disturbance, gaseous C other than $CO_2$ |
| Soils $CSE_{soil}$ | Litterfall and rhizodeposits ($\approx$NPP) | Net soil C balance | Through-fall | Heterotrophic respiration | - | - | Leaching, erosion |
| Sediments $CSE_{sediment}$ | NPP ($\approx$0 in deep sediments) | Net sediment C balance | Deposition | Heterotrophic respiration | - | - | - |
| Ecosystems and watersheds $CUE_{ecosystem}$ (or $CSE_{ecosystem}$) | Gross primary productivity | Net ecosystem productivity | Lateral C inputs (CSE: deposition) | Autotrophic and heterotrophic respiration | - | - | Gaseous C other than $CO_2$ (CSE: leaching, erosion) |


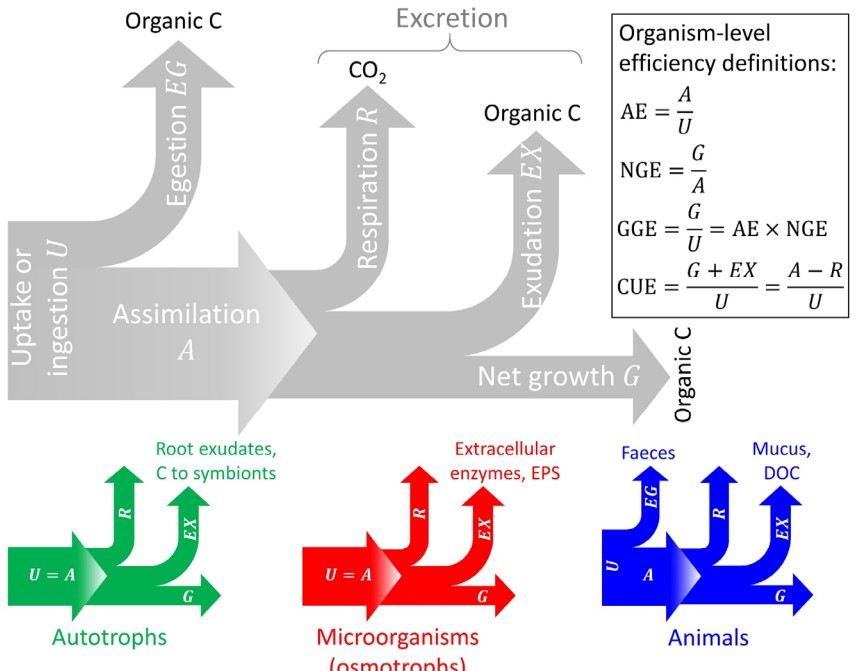

**Figure 1. General conceptual summary of C exchanges of individual organisms (or populations). Top: general terminology and C-based efficiency definitions (Sterner and Elser, 2002). Bottom: C exchanges of autotrophs, microorganisms feeding through the cell membranes (osmotrophs), and animals; note that assimilation is equal to uptake (or ingestion) in autotrophs and osmotrophs that lack guts, so that egestion cannot occur. The type of excretion product is also indicated (EPS: extracellular polysaccharides). Colour codes for the different organisms are used also in other figures.**


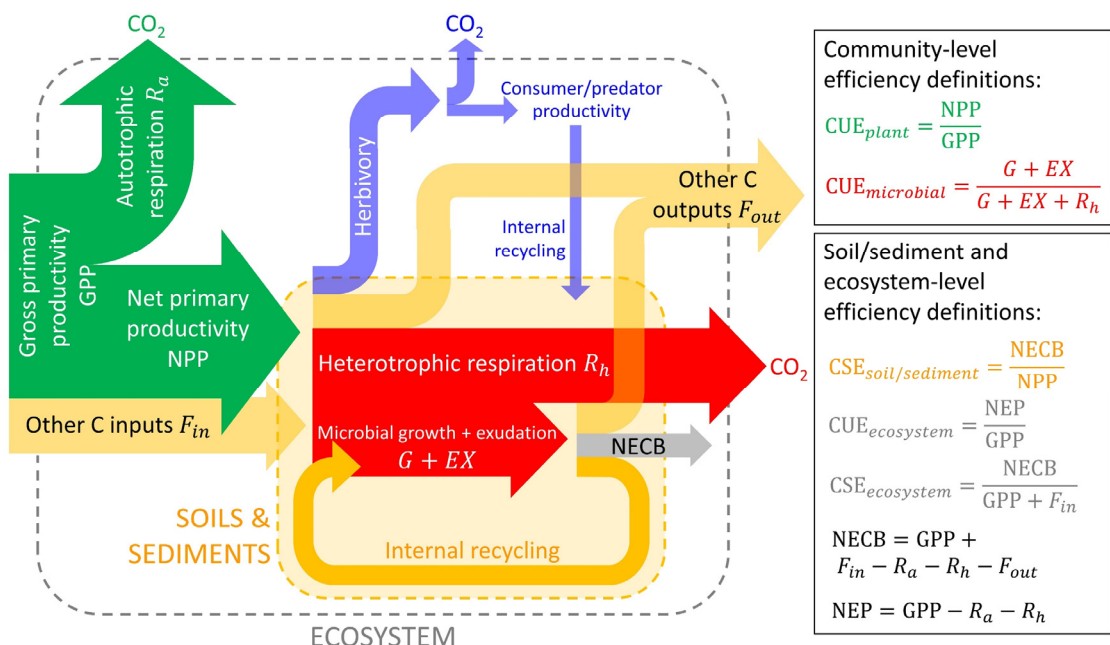


**Figure 2. Conceptual scheme of C fluxes in a generic ecosystem, following the terminology by Chapin et al. (2006), but adding the herbivory pathway. The ratio between the flux of C retained in a given sub-system (e.g., vegetation in green, microbial biomass in red, soil and sediments in yellow, whole ecosystem in grey) over the C flux taken up represents the C-use or C-storage efficiencies (CUE or CSE) of that sub-system. The net ecosystem C balance is denoted by NECB**

**and the net ecosystem productivity by NEP (not shown). C transport processes and C exchanges in forms other than $CO_2$ are denoted 'Other C inputs/outputs'. Colour codes for the different organisms and sub-systems are used also in other figures.**

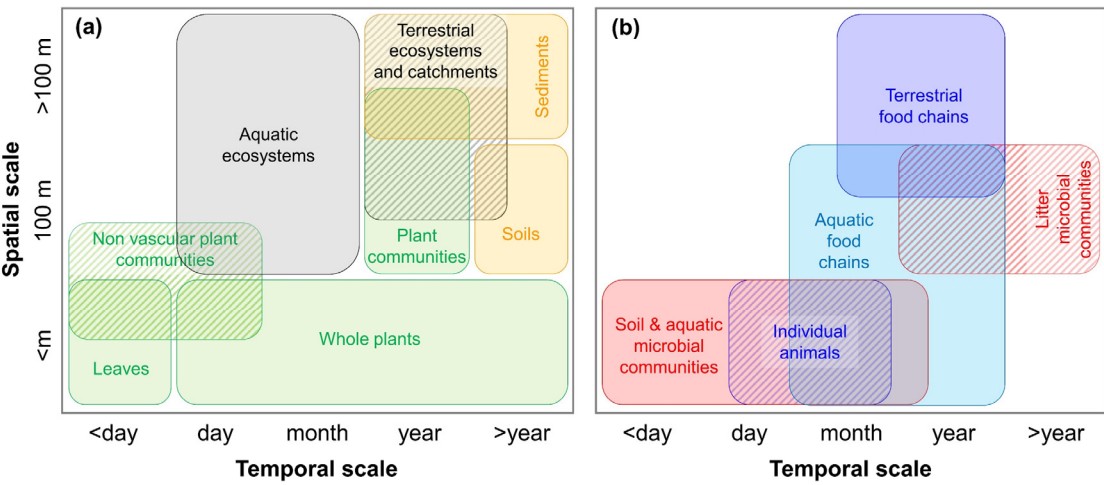

Figure 3. Illustration of typical spatial and temporal scales at which C fluxes are calculated to estimate CUE (or CSE) in various sub-systems. a) Scales typical of observations on vegetation, whole ecosystems, and soils/sediments; b) scales typical of observations on heterotrophic organisms and food chains. Colour codes are as in Fig. 2.

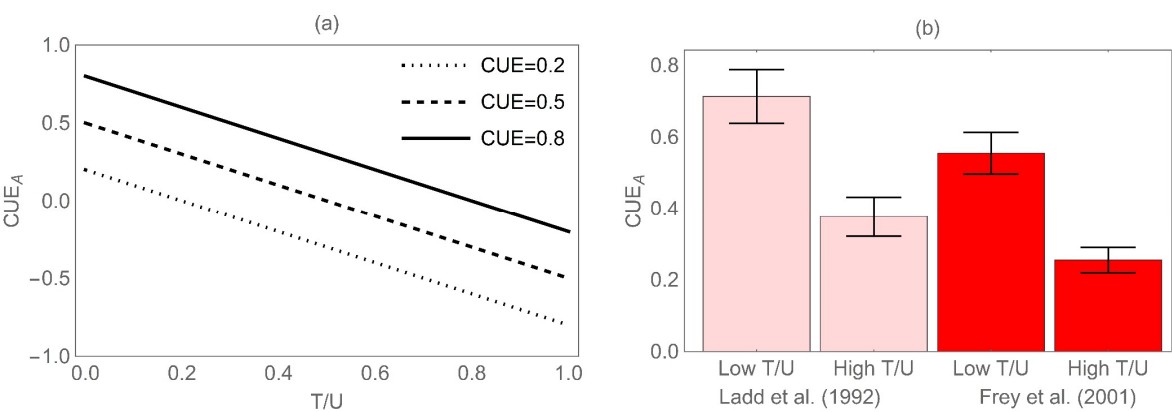


Figure 4. Effect of biomass turnover rate on the apparent C-use efficiency ($CUE_A$). (a) Theoretical relation between $CUE_A$ and the ratio of turnover rate over C uptake rate (Eq. (8), assuming negligible *EX*), for three values of the actual CUE. (b) Two examples of how high turnover rates cause a decrease in $CUE_A$ in empirical studies on soil microbial communities (Frey et al., 2001; Ladd et al., 1992). Lower turnover rates were caused by lower mortality in the first 3 days of incubation compared to the day 112 (Ladd et al., 1992), or by lower grazing in the first two days of incubation compared to days 7-8 (Frey et al., 2001). Error bars indicate standard errors of the mean (variability is across three soil types in Ladd et al. (1992) and across replicates and soil types in Frey et al. (2001)).


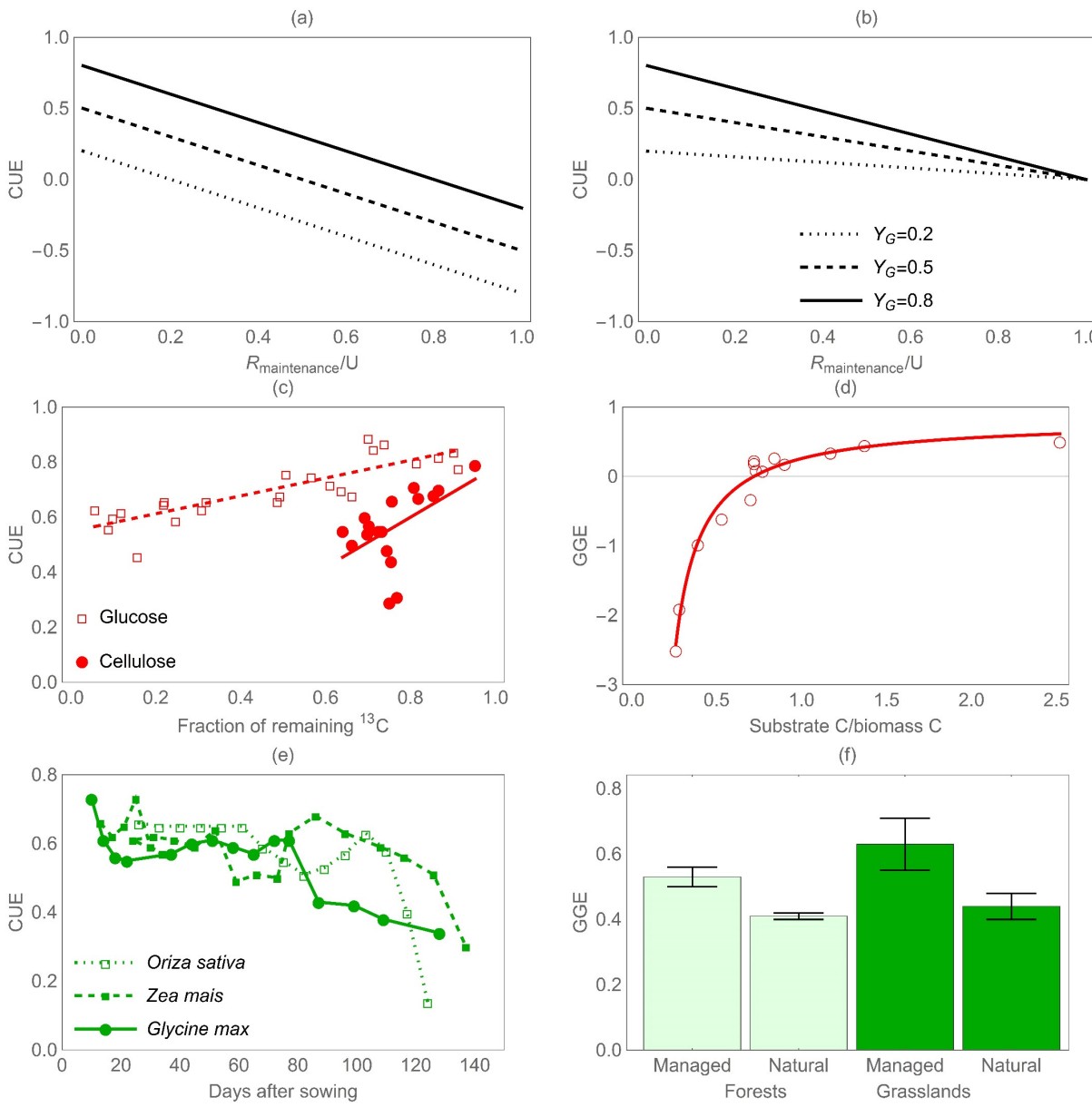


**Figure 5. Effect of maintenance respiration ($R_{maintenance}$) on C-use efficiency (CUE). Theoretical relations between CUE and the ratio of maintenance respiration over C uptake rate under two different assumptions: (a) priority to growth respiration, and (b) priority to maintenance respiration, for three values of growth yield (i.e., (C uptake - growth respiration)/C uptake). The central panels show decreasing CUE when (c) the C substrate is consumed (moving right**

**to left along the abscissa) during 12 (glucose) and 71 (cellulose) day incubations (Öquist et al., 2017) or (d) resource availability (as the ratio of salicylic acid C to biomass C) is low (Collado et al., 2014). (e) Reduction in CUE through time, as plants end their growth phase and set seeds (Yamaguchi, 1978). (f) Significantly higher ($p$<0.05) GGE of managed, and thus more nutrient-rich, forests and grasslands (Campioli et al., 2015). In (c) to (e), CUE or GGE decrease as costs for maintenance respiration increase relative to growth respiration; in (f), GGE decrease when costs**

**for symbiotic associations are higher (natural systems). Curves in (c) and (d) are least square linear and hyperbolic regressions drawn to guide the eye; error bars indicate standard errors of the mean.**

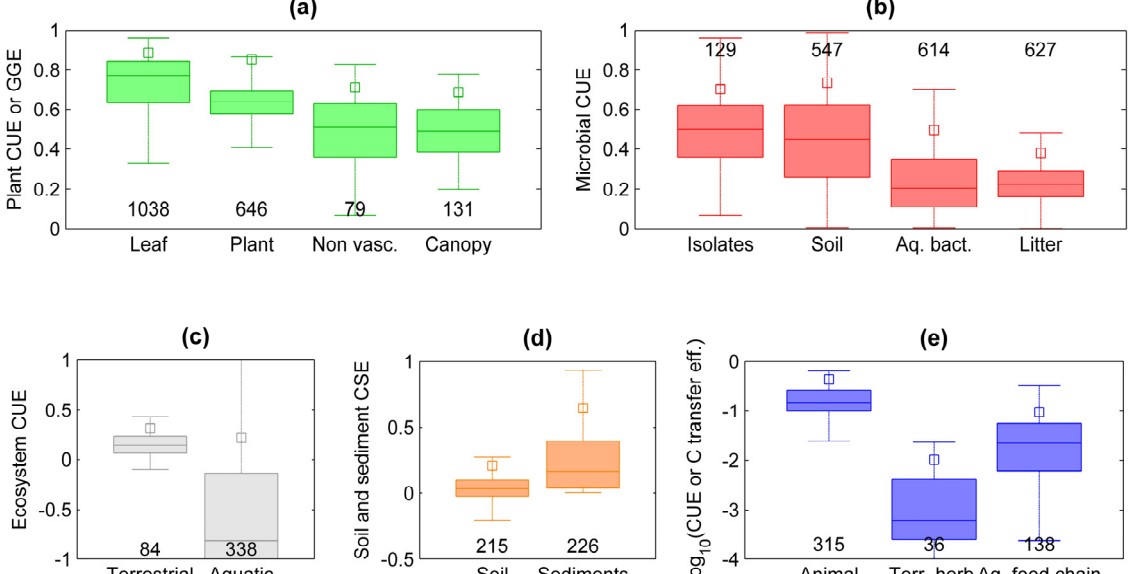

**Figure 6. Patterns in C-use efficiency (CUE) across scales and levels of organization. (a) CUE of leaves and non-vascular plant communities, and GGE of whole plants and vascular plant communities; (b) CUE of microbial isolates, and communities of soil microorganisms, aquatic bacteria, and litter microorganisms; (c) CUE of terrestrial and aquatic ecosystems (note that the y-axis extends to -1, indicating C losses larger than primary productivity); (d) C-storage efficiency (CSE) of soils and sediments (note that the y-axis extends to -0.5); (e) CUE of individual animals and terrestrial herbivore communities, and C transfer efficiency of aquatic food chains, plotted on a log-scale to allow a visual comparison. The box plots display median and quartiles (box), range excluding outliers (whiskers), and the open squares indicate the 90th percentiles; numbers indicate sample sizes; colour codes are as in Fig. 2. Data sources are described in the Supplementary Information.**

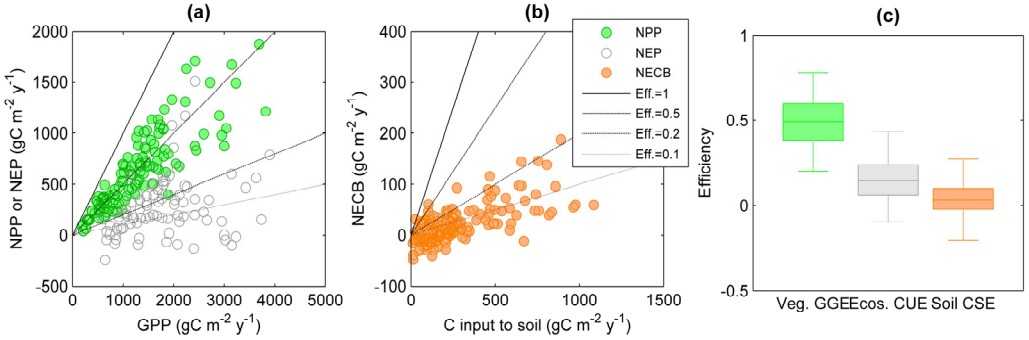

**Figure 7. Relations between gross primary productivity (GPP) and (a) net primary productivity (NPP) or net ecosystem productivity (NEP), and (b) between NPP and net ecosystem C balance (NECB) in terrestrial systems. In these plots, CUE corresponds to the slopes of lines passing through the origin (four are shown for illustration). (c) Comparison of vegetation GGE, whole ecosystem CUE, and soils CSE (see also Fig. 2). Colour codes are as in Fig. 2. Data sources are described in the Supplementary Information.**

