# Peer review of "Reviews and syntheses: Carbon use efficiency from organisms to ecosystems – Definitions, theories, and empirical evidence"

_Biogeosciences, 2018_

## Referee Comment (RC1) · Anonymous Referee #1 · 25 Jun 2018

Manzoni et al reviewed and synthesized patterns in carbon use efficiency (CUE) across scales. This is a large effort that can help reconciling previously identified differences in CUE. The authors go into the details of the different definitions that have been used and clarify some of the misunderstandings in the past. I think this could become an important contribution to the field, as differences in definitions and equations for CUE have been mostly ignored and confusion exists on what CUE should reflect. However, I do not fully agree on the presented definitions and think the manuscript still fails to fully resolve discrepancies. The current manuscript does not accurately represented what CUE is, where the term originates from and how it has been used in the past. As the manuscript reads now, I find it a missed opportunity to resolve the confusion that is

associated with this topic.

About the definitions:

From a plant perspective, theory indicates that CUE = NPP/GPP, with NPP = the synthesis of organic compounds = GPP – R. Hence, CUE = 1-R/GPP. (NPP = net primary production; GPP = gross primary production, R = respiration)

This corresponds more or less to equation 3 (CUE = 1-outputs/inputs) used by Manzoni et al. However, Manzoni et al consider egestion (EG) and exudation (EX, including symbionts) as part of the outputs, and not as part of NPP. Consequently, the CUE considered here is actually biomass production efficiency (biomass production/inputs = BPE) instead of CUE. Both CUE and BPE are meaningful terms – CUE focusses on the C cycle, while BPE targets the biomass that is produced. In the past, both terms have been rarely distinguished though and they have also not been used consistently. Correct use can be critical, however, as CUE and BPE may respond differently to environmental changes. For example, an increase in BPE could be associated with unaltered CUE if the partitioning to EX is the sole responsible of the change in BPE (i.e., R unchanged). Such understanding becomes important for example when comparing models with observations. Model evaluation assuming observed BPE = modeled CUE (=1-R/GPP) can lead to serious flaws, as illustrated by the following hypothetical example. Assuming modeled CUE should equal observed BPE, a (hypothetical) decrease of BPE with increasing CO2 concentration would suggest and increase in R/GPP whereas in reality the decrease in BPE may be solely due to an increase of EX while R/GPP, and hence true CUE, remain unaltered. In this hypothetical example, adjusting the model to reflect observed BPE in modeled CUE would lead to an overestimation of the response of CUE and R to elevated CO2.

The above problem related to the assumption that BPE = CUE is more prominent at some levels (e.g. vegetation) than at others (e.g. bacteria). Hence, differences among levels may in part be due to differences in the definition used. This is somewhat

acknowledged by the authors, but it would be much clearer and more accurate if BPE and CUE were clearly distinguished throughout the manuscript and if it was made clear in the figures and tables where BPE is calculated, where CUE is calculated and perhaps also where BPE $\sim$ CUE.

Specific comments: l.26 and l.50: I don't think biomass production/C uptake is the consensus definition of CUE (see above). Intro: I suggest to review the history of the definitions for CUE more elaborately. Where was it first used, what was the exact definition, how have definitions been applied in different fields,...? l. 160: clearly define the difference between uptake and assimilation to help the reader in following the different equations l.175: define overflow respiration. Ion uptake respiration is not mentioned. Is it considered part of growth respiration? See for example Lambers et al 1983, Physiologia Plantarum, 58: 556-563. l.200: replace 'reduces' with 'can be simplified to'. l.222: add 'and to EX' after 'exports to other parts of the plant'. l.295: I suggest to replace 'lower estimates of CUE' by 'an underestimation of CUE'. l.444: 'for a given uptake rate' seems more logical than 'for a given respiration rate'.

I think the authors missed some relevant publications. Cotrufo et al 2013 (Global Change Biology 19, 988-995) discuss the influence of substrate quality on microbial substrate use efficiency (another alternative for CUE), and consequences for soil C storage. This framework deserves at least a mention. Campioli et al 2015 (Nature Geoscience 8, 843-846) provide an update of Luyssaert et al 2007 and Vicca et al 2012 (both cited in the manuscript), and include also other vegetation types than forests. Data are provided in the supplementary files. I suggest considering including these data, or at least refer to them.

Table 2: Cannell and Thornley 2000 actually used the definition CUE = 1- Ra/GPP. DeLucia et al 2007 used data on biomass production/GPP but termed it NPP/GPP (hence ignoring other NPP components such as exudates and symbionts). This is part of the confusion and I suggest the authors take the opportunity to clarify this. Fig. 2: CUEplant is defined as NPP/GPP, but NPP is undefined. In line with my earlier

comments, I suggest to clearly define NPP. Figs. 6 and 7: clarify where the data originate from (refer to SI).

---

## Referee Comment (RC2) · Anonymous Referee #2 · 5 Jul 2018

The manuscript submitted by Manzoni et al. is a review associated to a database analyse around the concept of carbon use efficiency and carbon storage efficiency. The quality of the manuscript is very high and I particularly appreciated the effort of the authors to gather data from very different sources to have a broad view of the CUE/CSE concept. The writing is excellent and despite the complexity of the question the authors succeed to make a clear and easy to read document. I am convinced that this paper will be provide an important contribution to the literature and since it deals with data coming from plants, soil, ocean, etc. at different spatiotemporal scales it is of broad interest.

I may have few minor comments to try to make the manuscript even more attractive.

Section 3. Can you provide a bit more details on the methods used to collect the data (e.g. keywords used in ISIWEB).

Section 4.1 You cite two studies as example but some methodological details are missing to fully understand your arguments (what kind of carbon added (litter, glucose, ...) or how long was the incubation for instance).

I missed some words on the anthropogenic effect on ecosystems CSE. In all the manuscript you compared different types of ecosystems but it may be interesting to compare systems highly managed like cropland or European forest and grassland with a substantial fraction of the NPP appropriated by humans (see Krausmann et al. (2013) for instance).

As a modeller I have a very selfish request (but I guess it may help others). I appreciated the section 4.5 but I guess that the majority of the modellers using CUE concept are aware of the limitations presented here. Maybe one or two paragraphs with some concrete recommendations will be helpful. In particular, I am wondering if CUE or CSE at organisms or ecosystem levels should be considered as emerging properties of a given system and if yes it might become an interesting approach to evaluate model by comparing the CUE/CSE observed at the system level.

Krausmann, F., Erb, K.-H., Gingrich, S., Haberl, H., Bondeau, A., Gaube, V., Lauk, C., Plutzar, C. and Searchinger, T. D.: Global human appropriation of net primary production doubled in the 20th century, Proc. Natl. Acad. Sci., 110(25), 10324–10329, doi:10.1073/pnas.1211349110, 2013.

---

## Referee Comment (RC3) · Anonymous Referee #3 · 13 Jul 2018

The manuscript is descriptive without a very extensive data analysis. However, the synthesis is new (I've never read about such large comparison of CUE across biological systems and biological scales) and interesting (I particularly like the fundamental Fig. 6). So, I think the manuscript is suited for publication without a data re-analysis.

However, there are key points that need to be improved (do not underestimate them, even are just text improvements). The Theory (paragraph 2) and definitions are fundamental in this paper, yet are not fully clear.

*for all biological systems, you use the term CUE. However, as well reported in Table 2, for some systems other terms are used. Furthermore, CUE is associated to a specific

variable/system (plant and community CUE=NPP/GPP). It would have been much less confusing (and more relevant) if you were proposing an overarching (new) efficiency term, and not 'impose' the one used for some systems to all cases.

*your attempt of generalization (paragraph 2) is not always easy to follow because each domain (plant, micro-organisms, ecosystems etc.) has is own specific definitions and terminology. It would be easier if you, before generalize (so before paragraph 2.1), describe the specific ways CUE is calculated for each of the five 'scales' you synthesize in Fig 6, thus an extension of Table 2. And then, when you generalize, make several examples. For instance, what is 'Output' (Eq. 1) for the five scales?

*There are the definitions used in the field-specific literature (Table 2) and you add other definitions: CUEapparent, AE, NGE, GGE, CUE ecosystem (extremely confusing: NPP/GPP or NEP/GPP?). Make some choices (can the definitions be reduced?) and clarify.

*For some cases, you mention the possibility of negative CUE, but for plant (CUE=NPP/GPP) it would not be possible because NPP>0 or =0). Similarly, turnover has a meaning for microbes and another for plants (e.g. in forests, turnover refers to the annual leaves, branch or root turnover and it is added in NPP, Clark et al 2001 Ecological Application 11(2), pp. 356–370).

Other remarks

*Your main key syntheses were (from abstract): (i) CUE increases with improving growing conditions, (ii) CUE decrease due to turnover, (iii) CUE decreases with increasing biological and ecological organization. Write them also in Conclusions (instead of generic sentences from L497 to L505) with the key reasons/explanations.

*L320-321: as in plants CUE= NPP/GPP and seed production is accounted in NPP, I do not understand your point . . .

*L503 can be move above where you discuss applicability of CUE values.

\*You do not make reference to Campioli et al 2015 Nat Geo. However, that synthesis can be useful, not only for the additional dataset on CUE (that they consider BPE there) but for comparison of ecosystems of different complexity (e.g. natural grassland vs. cropland monoculture). Also there are various suggestions for practical use of CUE/BPE in that paper.
* * *

---

## Author Comment (AC1) · 16 Aug 2018

**General Response to Referees' Comments**

Three anonymous referees have provided generally positive comments on our manuscript, with some constructive critiques that we plan to address as explained below (responses are written in *italic*; modified text in quotation marks). The three main comments are as follows:

- Reviewer #1: ambiguous definition of C-use efficiency (CUE) for organisms and communities. The reviewer correctly argues that CUE should be defined as 1-respiration rate/uptake rate. This means that biological products such as exudates should be accounted for in the CUE definition, differently from gross growth or biomass production efficiency in which net biomass increments only are accounted for. We will revise our definition according to the reviewer's suggestion.
- Reviewer #2: lacking discussion on human impacts on ecosystem scale efficiencies. This is an excellent suggestion as it prompted us to generalize the definition of C storage efficiency (CSE). We will include anthropogenic C fluxes in our definition of CSE and comment on the role of ecosystem management as a driver of change for CSE.
- Reviewer #3: lack of clarity in some definitions. We will revise the text to clarify ambiguous definitions, and change Table 3 to indicate which components of the C cycle represent inputs and outputs in each of the systems/scales considered.

Detailed responses are reported in the individual rebuttal letters.

**Anonymous Referee #1**

Manzoni et al reviewed and synthesized patterns in carbon use efficiency (CUE) across scales. This is a large effort that can help reconciling previously identified differences in CUE. The authors go into the details of the different definitions that have been used and clarify some of the misunderstandings in the past. I think this could become an important contribution to the field, as differences in definitions and equations for CUE have been mostly ignored and confusion exists on what CUE should reflect. However, I do not fully agree on the presented definitions and think the manuscript still fails to fully resolve discrepancies. The current manuscript does not accurately represented what CUE is, where the term originates from and how it has been used in the past. As the manuscript reads now, I find it a missed opportunity to resolve the confusion that is associated with this topic.

*We agree that the manuscript can be improved along the lines suggested below and we appreciate the reviewer's efforts to exemplify his/her concerns. This is a very helpful evaluation that led to a more rigorous definition of CUE vs. BPE terms.*

About the definitions:
From a plant perspective, theory indicates that CUE = NPP/GPP, with NPP = the synthesis of organic compounds = GPP – R. Hence, CUE = 1-R/GPP. (NPP = net primary production; GPP = gross primary production, R = respiration). This corresponds more or less to equation 3 (CUE = 1-outputs/inputs) used by Manzoni et al. However, Manzoni et al consider egestion (EG) and exudation (EX, including symbionts) as part of the outputs, and not as part of NPP. Consequently, the CUE considered here is actually biomass production efficiency (biomass production/inputs = BPE) instead of CUE. Both CUE and BPE are meaningful terms – CUE focusses on the C cycle, while BPE targets the biomass that is produced. In the past, both terms have been rarely distinguished though and they have also not been used consistently.

*We agree that this is a critical point. In the original manuscript, we had noted this discrepancy between BPE and CUE and pointed to their different interpretations. The two definitions differ if egestion and exudation are important components, as in the case of animals (mainly egestion) and plants (mainly exudation). Whether the distinction matters for microbial community is debatable, as estimates of exudation rates in natural communities are not available. While we agree fully on the importance of the distinction, the literature does not agree on how to call these different efficiencies across disciplines. For example, classical ecological stoichiometry texts (Sterner and Elser 2002) use the term 'gross growth efficiency' (GGE) instead of 'biomass production efficiency' (BPE), which has become common in papers focusing on stand-scale carbon balances. The same term CUE indicates – depending on the source – different efficiencies (including exudation or not), creating some confusion. Therefore, an unambiguous definition of CUE and an explanation on how it differs from GGE (=BPE) are indeed much needed.*

Correct use can be critical, however, as CUE and BPE may respond differently to environmental changes. For example, an increase in BPE could be associated with un-altered CUE if the partitioning to EX is the sole responsible of the change in BPE (i.e. R unchanged). Such understanding becomes important for example when comparing models with observations. Model evaluation assuming observed BPE = modelled CUE (=1-R/GPP) can lead to serious flaws, as illustrated by the following hypothetical example. Assuming modeled CUE should equal observed BPE, a (hypothetical) decrease of BPE with increasing $CO_2$ concentration would suggest and increase in R/GPP whereas in reality the decrease in BPE

may be solely due to an increase of EX while R/GPP, and hence true CUE, remain unaltered. In this hypothetical example, adjusting the model to reflect observed BPE in modeled CUE would lead to an overestimation of the response of CUE and R to elevated CO2.

*This is a very useful example, and we fully agree that there is a risk of misinterpretation. We will revise the text to clarify this issue, as described below.*

The above problem related to the assumption that BPE = CUE is more prominent at some levels (e.g. vegetation) than at others (e.g. bacteria). Hence, differences among levels may in part be due to differences in the definition used. This is somewhat acknowledged by the authors, but it would be much clearer and more accurate if BPE and CUE were clearly distinguished throughout the manuscript and if it was made clear in the figures and tables where BPE is calculated, where CUE is calculated and perhaps also where BPE~CUE.

*We will implement a clearer definition of both – CUE and BPE. For 'historical' reasons, we will use the term GGE (Sterner and Elser 2002), but making sure to underline that BPE is a completely equivalent term. Regarding CUE, we will adopt the suggested definition of 1-respiration/uptake – that is, CUE is a measure of C conversion to organic compounds that will eventually become new biomass or will be exuded (plants, microbes). When changing this definition, we will clarify that it differs from the one proposed earlier by Manzoni et al. (2012), but that it is preferred for consistency with more recent work, especially in the areas of plant physiology and in stand-scale C cycling studies. Using the notation of Sterner and Elser (2002), CUE is thus defined as:*

*CUE = (G + EX)/U = G/U + EX/U = GGE + EX/U.*

*Using this relation allows explaining how CUE~GGE (or equivalently CUE~BPE) only when EX<<U. Given the proliferation of terms with different meaning, we will also add a comparison table in the Supplementary Information, in addition to the current Tables 2 and 3:*

| Definition | Context | Originally used terms | Source |
|---|---|---|---|
| $CUE_A$ = 1-O/I | Soil microbial communities | Ecosystem-scale efficiency of microbial biomass synthesis and recycling of necromass/exudates ($CUE_E$) | (Eq. 2 in Geyer et al. 2016) |
| GGE = G/U | Animals and microorganisms | Gross growth efficiency (GGE) | (Sterner and Elser 2002) |
| | Microbial communities | Carbon use efficiency (CUE) | (Eq. 2 in Manzoni et al. 2012) |
| | Soil microbial communities | Community-scale efficiency of microbial biomass synthesis ($CUE_C$) | (Eq. 1 in Geyer et al. 2016) |
| | Individual plants | Carbon use efficiency (CUE) | (Gifford 1995) |
| | Plant communities | Biomass production efficiency (BPE) | (Campioli et al. 2015) |
| CUE = 1-R/U | Soil microbial communities | Community-scale efficiency of microbial biomass synthesis when EX≈0 (also denoted as $CUE_C$) | (Figure 3 in Geyer et al. 2016) |
| | Plant communities | Carbon use efficiency (CUE = NPP/GPP) | (Cannell and Thornley 2000) |

*There are also many more other definitions of growth/uptake ratios for both heterotrophs and autotrophs, but their interpretation depends on the measurements done to estimate growth and uptake rates (see Table 2 in the main text). In the large majority of cases, the exudation rate and its role in differentiating between GGE and CUE is not mentioned.*

*Moreover, with the definition CUE = 1-R/U, C losses from organisms and ecosystems are treated in the same way –in both cases CUE is defined as the amount of C that is fixed in organic form and not respired. Exudation and leaching (for organisms and ecosystems, respectively) are subtracted from this quantity and lead to the definition of GGE or net ecosystem C balance (NECB). This perspective will be also presented in the revised manuscript.*

*To summarize, in response to this important remark, we will amend the text on the definition of organism- and community-level CUE as follows (we will also update the Supplementary Information along the same lines):*

*"We now define CUE at the organism level as the ratio between the rate of production of biomass and products (G+EX), and the rate of C uptake (U),*

$$CUE = \frac{G+EX}{U} = 1 - \frac{EG+R}{U}. \tag{5}$$

*As a result, the mass balance equation can be rewritten as,*

$$\frac{dC}{dt} = CUE \times U - EX - T. \tag{6}$$

*With this definition, CUE represents the fraction of C taken up allocated to biosynthesis (biomass and products that will be eventually exuded), but excluding respired and egested C, which do not contribute to biosynthesis. Including exudates such as enzymes and polymeric compounds in the CUE definition is motivated by the clear fitness advantage these products have for the organism. Moreover, C storage compounds and osmolytes are also regarded as 'biomass', as they would be measured as cellular material.*

*This definition is consistent with previous work on plant C budgets (Thornley and Cannell 2000), but it differs from definitions often used for soil microorganisms where only biomass synthesis is considered and CUE = GGE (Manzoni et al. 2012, Geyer et al. 2016). It is thus important to emphasize that CUE is higher than GGE. The difference between GGE and CUE is relevant when EX is large, as in the case of organic C exchanges between roots and plant symbionts (Hobbie 2006, Ekblad et al. 2013). For microbial communities, the entity of the extracellular enzyme and polysaccharide synthesis is unknown but presumably small compared to the other rates involved, at least in soils (Schimel and Weintraub 2003), making the distinction less important in practice (for further discussions in this context, see Geyer et al. 2016)."*

Specific comments: l.26 and l.50: I don't think biomass production/C uptake is the consensus definition of CUE (see above). Intro: I suggest to review the history of the definitions for CUE more elaborately. Where was it first used, what was the exact definition, how have definitions been applied in different fields...

*In most fields, CUE is defined as a biomass production over uptake, but we agree that in the case of plants, CUE is defined as NPP/GPP. The term CUE was first used by plant physiologists in the early 1990's based on a search with ISI Web of Science. We cite for example one of those early papers, in which CUE is actually defined as net biomass production over gross photosynthesis (Gifford 1995). That initial definition is thus not consistent with the one proposed by the reviewer. However, the same author changed the definition in his more recent work to CUE = NPP/GPP (Gifford 2003). Notably, the term "carbon use efficiency" was also used in the same period and discipline (plant physiology), but with a completely different meaning – the net C gain in terms of plant biomass per unit C spent to support mycorrhizal associations (Tinker et al. 1994). Therefore, even within a specific discipline, there is no consensus on the use of these terms.*

*We will provide a short historical overview of the efficiencies of C conversion, to give readers an idea of how terms have evolved, but we will not cover the historical angle comprehensively, as it would require a lengthy presentation of terms for all the disciplines involved. Table 2 in the main text already contains key publications (with early papers from the 1970's), and we will add further key references such as the seminal paper by Monod (1949). The interested reader can follow the evolution of the terminology starting from those references. However, we will take the opportunity offered by the historical context for introducing the main difficulties in defining and interpreting CUE, and change the second paragraph of the Introduction as follows:*

*"For biological systems (organs, individual organisms, or even entire communities), CUE is defined as the ratio between the amount of C allocated to biosynthesis (new biomass and biological products, including e.g., exudates) and the amount of C taken up. While the term CUE was proposed in the mid-1990s in the context of plant C balances (Gifford 1995), other terms – e.g., 'growth yield' – referring to the efficiency of substrate conversion into biomass had been in use since the early 1900 (Monod 1949). Now, efficiency definitions are proliferating across many disciplines in biology, ecology, and Earth sciences. While some of these definitions are comparable (and all are deceptively simple), subtle differences often emerge, partly due to conceptual and methodological advances that allow quantification of previously ignored C exchanges. These differences and in general the evolution of CUE definitions make interpretation of results difficult and complicate cross-disciplinary comparisons.*

*The main difficulty is to unambiguously define what represents growth or C storage, and reconcile conceptual definitions with empirical estimates (Clark et al. 2001, Chapin et al. 2006, Geyer et al. 2016)..."*

l. 160: clearly define the difference between uptake and assimilation to help the reader in following the different equations

*We will change the sentence below Eq. 4 as "where U is the uptake rate, U-EG is the assimilation rate (i.e., A in Fig. 1), and G is the net growth rate in C units".*

l.175: define overflow respiration. Ion uptake respiration is not mentioned. Is it considered part of growth respiration? See for example Lambers et al 1983, Physiologia Plantarum, 58: 556-563.

*Quoting the more recent review by Cannell and Thornley (2000), "Ion uptake and phloem loading have elements of both growth and maintenance and all forms of wastage respiration are neither"(p. 49). We have indeed separated wastage (overflow) respiration from the other components, but we would prefer not to delve into the discussion on whether ion uptake should be included in growth or maintenance respiration. This issue is important in practice for modelling purposes, but we feel it is outside our goals in this contribution. However, we will clarify that overflow respiration includes wastage respiration and any C respired to compensate stoichiometric imbalances.*

l.200: replace 'reduces' with 'can be simplified to'.
l.222: add 'and to EX' after 'exports to other parts of the plant'.
l.295: I suggest to replace 'lower estimates of CUE' by 'an underestimation of CUE'.
l.444: 'for a given uptake rate' seems more logical than 'for a given respiration rate'.

*Thanks for these comments – we will implement the suggested changes*

I think the authors missed some relevant publications. Cotrufo et al 2013 (Global Change Biology 19, 988-995) discuss the influence of substrate quality on microbial substrate use efficiency (another alternative for CUE), and consequences for soil C storage. This framework deserves at least a mention.

*We will add the suggested reference and include a comment on recent results of positive correlation between microbial CUE and soil C storage (Kallenbach et al. 2016) – consistent with the framework proposed by Cotrufo et al. (2013).*

Campioli et al 2015 (Nature Geoscience 8, 843-846) provide an update of Luyssaert et al 2007 and Vicca et al 2012 (both cited in the manuscript), and include also other vegetation types than forests. Data are provided in the supplementary files. I suggest considering including these data, or at least refer to them.

*The dataset by Campioli et al. (2015) provides a new interpretation of nutrient availability effects on forest BPE, which we will also present in the revised manuscript. After checking the dataset itself, it appears to be less useful than the Luyssaert et al. (2007) dataset for the purpose of comparing plant community and ecosystem CUE because it focuses on GPP and biomass production. Therefore, we do not plan to change the analysis made for Figures 6 and 7 in the main text. Please see also the response to Reviewer 2 regarding the role of human activities on ecosystem CUE and CSE – in the new Discussion paragraph, we will cite the paper by Campioli et al. (2015).*

Table 2: Cannell and Thornley 2000 actually used the definition CUE = 1- Ra/GPP.

*That is correct. We will move the citation to Cannell and Thornley (2000) to the correct position in the table, next to DeLucia et al. (2007), and cite here Gifford (1995) instead.*

DeLucia et al 2007 used data on biomass production/GPP but termed it NPP/GPP (hence ignoring other NPP components such as exudates and symbionts). This is part of the confusion and I suggest the authors take the opportunity to clarify this.

*We will add a clarification to this regard in the Supplementary information (Section 1.3):*

*"As shown in Eq. (4) in the main text, plant community CUE should be calculated by including all these C fluxes in the numerator. When only net biomass increments are available, the terms gross growth efficiency (GGE) or biomass production efficiency (BPE) are more accurate (as in Vicca et al. 2012, Campioli et al. 2015)."*

Fig. 2: CUEplant is defined as NPP/GPP, but NPP is undefined. In line with my earlier comments, I suggest to clearly define NPP.

*We will more clearly define NPP in the main text and in the SI (Section 1.3), but would prefer not to add more definitions in the caption, in which we already refer to Chapin et al. (2006), where NPP is also defined.*

Figs. 6 and 7: clarify where the data originate from (refer to SI)

*We will refer to the SI by adding: "Data sources are described in the Supplementary Information."*

**References**

Campioli, M., S. Vicca, S. Luyssaert, J. Bilcke, E. Ceschia, F. S. Chapin Iii, P. Ciais, M. Fernandez-Martinez, Y. Malhi, M. Obersteiner, D. Olefeldt, D. Papale, S. L. Piao, J. Penuelas, P. F. Sullivan, X. Wang, T. Zenone, and I. A. Janssens. 2015. Biomass production efficiency controlled by management in temperate and boreal ecosystems. Nature Geosci **8**:843-846.

Cannell, M. G. R., and J. H. M. Thornley. 2000. Modelling the components of plant respiration: Some guiding principles. Annals of Botany **85**:45-54.

Chapin, F. S., G. M. Woodwell, J. T. Randerson, E. B. Rastetter, G. M. Lovett, D. D. Baldocchi, D. A. Clark, M. E. Harmon, D. S. Schimel, R. Valentini, C. Wirth, J. D. Aber, J. J. Cole, M. L. Goulden, J. W. Harden, M. Heimann, R. W. Howarth, P. A. Matson, A. D. McGuire, J. M. Melillo, H. A. Mooney, J. C. Neff, R. A. Houghton, M. L. Pace, M. G. Ryan, S. W. Running, O. E. Sala, W. H. Schlesinger, and E. D. Schulze. 2006. Reconciling carbon-cycle concepts, terminology, and methods. Ecosystems **9**:1041-1050.

Clark, D. A., S. Brown, D. W. Kicklighter, J. Q. Chambers, J. R. Thomlinson, and J. Ni. 2001. Measuring net primary production in forests: Concepts and field methods. Ecological Applications **11**:356-370.

Ekblad, A., H. Wallander, D. L. Godbold, C. Cruz, D. Johnson, P. Baldrian, R. G. Bjork, D. Epron, B. Kieliszewska-Rokicka, R. Kjoller, H. Kraigher, E. Matzner, J. Neumann, and C. Plassard. 2013. The production and turnover of extramatrical mycelium of ectomycorrhizal fungi in forest soils: role in carbon cycling. Plant and Soil **366**:1-27.

Geyer, K. M., E. Kyker-Snowman, A. S. Grandy, and S. D. Frey. 2016. Microbial carbon use efficiency: accounting for population, community, and ecosystem-scale controls over the fate of metabolized organic matter. Biogeochemistry **127**:173-188.

Gifford, R. M. 1995. Whole plant respiration and photosynthesis of wheat under increased $CO_2$ concentration and temperature: Long-term vs short-term distinctions for modelling. Global Change Biology **1**:385-396.

Gifford, R. M. 2003. Plant respiration in productivity models: conceptualisation, representation and issues for global terrestrial carbon-cycle research. Functional Plant Biology **30**:171-186.

Hobbie, E. A. 2006. Carbon allocation to ectomycorrhizal fungi correlates with belowground allocation in culture studies. Ecology **87**:563-569.

Kallenbach, C. M., S. D. Frey, and A. S. Grandy. 2016. Direct evidence for microbial-derived soil organic matter formation and its ecophysiological controls. Nature communications **7**:13630.

Manzoni, S., P. G. Taylor, A. Richter, A. Porporato, and G. I. Ågren. 2012. Environmental and stoichiometric controls on microbial carbon-use efficiency in soils. New Phytologist **196**:79-91.

Monod, J. 1949. The growth of bacterial cultures. Annual Review of Microbiology **3**:371-394.

Schimel, J. P., and M. N. Weintraub. 2003. The implications of exoenzyme activity on microbial carbon and nitrogen limitation in soil: a theoretical model. Soil Biology & Biochemistry **35**:549-563.

Sterner, R. W., and J. J. Elser. 2002. Ecological stoichiometry. The biology of elements from molecules to the biosphere. Princeton University Press, Princeton and Oxford.

Thornley, J. H. M., and M. G. R. Cannell. 2000. Modelling the components of plant respiration: Representation and realism. Annals of Botany **85**:55-67.

Tinker, P. B., D. M. Durall, and M. D. Jones. 1994. Carbon use efficiency in mycorrhizas - Theory and sample calculations. New Phytologist **128**:115-122.

Vicca, S., S. Luyssaert, J. Penuelas, M. Campioli, F. S. Chapin, III, P. Ciais, A. Heinemeyer, P. Hogberg, W. L. Kutsch, B. E. Law, Y. Malhi, D. Papale, S. L. Piao, M. Reichstein, E. D. Schulze, and I. A. Janssens. 2012. Fertile forests produce biomass more efficiently. Ecology Letters **15**:520-526.

---

## Author Comment (AC2) · 16 Aug 2018

**General Response to Referees' Comments**

Three anonymous referees have provided generally positive comments on our manuscript, with some constructive critiques that we plan to address as explained below (responses are written in *italic*; modified text in quotation marks). The three main comments are as follows:

- Reviewer #1: ambiguous definition of C-use efficiency (CUE) for organisms and communities. The reviewer correctly argues that CUE should be defined as 1-respiration rate/uptake rate. This means that biological products such as exudates should be accounted for in the CUE definition, differently from gross growth or biomass production efficiency in which net biomass increments only are accounted for. We will revise our definition according to the reviewer's suggestion.
- Reviewer #2: lacking discussion on human impacts on ecosystem scale efficiencies. This is an excellent suggestion as it prompted us to generalize the definition of C storage efficiency (CSE). We will include anthropogenic C fluxes in our definition of CSE and comment on the role of ecosystem management as a driver of change for CSE.
- Reviewer #3: lack of clarity in some definitions. We will revise the text to clarify ambiguous definitions, and change Table 3 to indicate which components of the C cycle represent inputs and outputs in each of the systems/scales considered.

Detailed responses are reported in the individual rebuttal letters.

**Anonymous Referee #2**

The manuscript submitted by Manzoni et al. is a review associated to a database analyse around the concept of carbon use efficiency and carbon storage efficiency.

The quality of the manuscript is very high and I particularly appreciated the effort of the authors to gather data from very different sources to have a broad view of the CUE/CSE concept. The writing is excellent and despite the complexity of the question the authors succeed to make a clear and easy to read document. I am convinced that this paper will be provide an important contribution to the literature and since it deals with data coming from plants, soil, ocean, etc. at different spatiotemporal scales it is of broad interest.

I may have few minor comments to try to make the manuscript even more attractive.

*Thank you for the very supportive comments.*

Section 3. Can you provide a bit more details on the methods used to collect the data (e.g. keywords used in ISIWEB).

*Most of the datasets we used are already published. In this case we selected the most comprehensive published (and open-access) datasets that covered the widest range of ecosystems or organisms (see for details Table S1). We also collected specific data into new datasets to fill notable gaps. We aimed at having a representative sample of data, but did not perform a fully systematic data search as would be necessary in a meta-analysis, for two reasons:*

1) *our goal was to only illustrate general patterns and report simple statistics (median, variability, range), not to perform a fully quantitative analysis*
2) *the enormous variability in terminology used across disciplines, and the fact that in many cases C-use or C-storage efficiencies were not reported in the papers, but only the C fluxes to calculate them were provided, made a systematic search not feasible. We therefore often followed leads from key papers (highly cited or identified by one of the co-authors) to find a selection of publications covering a given angle of C cycling and reporting enough information to calculate efficiencies*

*For the new data compilations, we searched publications in the ISI Web of Science and in Google Scholar using various sets of keywords (refer to Table S1 for the following 'system' labels):*

*All plants: plant, "carbon use efficiency", growth AND respiration*

*Non-vascular plant communities: lichen OR bryophyte, respiration, temperature, "carbon use efficiency"; the compilation was based on existing synthesis papers (Porada et al. 2013, Lenhart et al. 2015)*

*Aquatic food chains: "food chain" OR "food web", "food chain efficiency" OR "carbon transfer efficiency"*

*Soils: soil, "organic carbon", "carbon sequestration efficiency", "carbon storage", "litter input" OR "residue input"*

*Catchments: "gross primary production", "dissolved organic carbon" OR "dissolved organic carbon export" OR "DOC leaching", "net ecosystem exchange", productivity, respiration*

*It should be noted that this was an iterative process, as we often started a search finding almost no relevant publication, only to later discover that in a specific sub-discipline different synonyms were used. Therefore, the search terms were updated accordingly, leading to a less structured search than we would have preferred. The result of this process is partly reflected by Table 2 in the main text, where all the synonyms found for CUE and CSE are listed.*

*To address this comment, we will add a paragraph in Section 3 "Data collection and analysis" that summarizes the above considerations:*

*"To compile the new data collections, we conducted an online search using ISI Web of Science and Google Scholar with keywords including various synonyms of CUE or CSE. We also gathered publications following relevant references in articles and books, and thanks to the expert knowledge of the authors. Due to the enormous variability in terminology used across disciplines, and the fact that in many cases CUE or CSE were not reported in the papers (but only C exchange rates to calculate them), a systematic search was not feasible. As a result, while not exhaustive, our selection of publications covers a broad range of conditions for each subset of data, and allows illustrating general patterns across disciplines and scales."*

Section 4.1 You cite two studies as example but some methodological details are missing to fully understand your arguments (what kind of carbon added (litter, glucose...) or how long was the incubation for instance).

*Details will be added in the main text to indicate the added compound used (glucose in both cases), and further clarifications will be added in the caption of Figure 4:*

*"Lower turnover rates were caused by lower mortality in the first 3 days of incubation compared to the day 112 (Ladd et al. 1992), or by lower grazing in the first two days of incubation compared to days 7-8 (Frey et al. 2001). Error bars indicate standard errors of the mean (variability is across three soil types in Ladd et al. (1992) and across replicates and soil types in Frey et al. (2001))."*

*We will also state for how long were the incubations performed to acquire the data shown in Figure 5c.*

I missed some words on the anthropogenic effect on ecosystems CSE. In all the manuscript you compared different types of ecosystems but it may be interesting to compare systems highly managed like cropland or European forest and grassland with a substantial fraction of the NPP appropriated by humans (see Krausmann et al. (2013) for instance).

*This is an excellent point. It is certainly possible to extend our definitions to systems that are heavily managed. We will first modify the definition of what are now referred to as "abiotic C exchanges", to accommodate anthropogenic intervention:*

*"The $F_{in}$ and $F_{out}$ are respectively C inputs and outputs occurring via abiotic exchanges of organic and inorganic C in natural ecosystems, but also account for anthropogenic inputs (e.g., manure) and outputs (e.g., harvested products) in managed ecosystems"*

*Second, we will refer more explicitly to anthropogenic C fluxes throughout the manuscript, and specifically add a paragraph in the discussion section 4.4 on human alterations of CSE:*

*"A large fraction of land surface and of marine systems is managed to extract food and fibre to support a growing human population (Krausmann et al. 2013). Management of ecosystems has two contrasting effects on CSE, depending on the balance of harvest removal, improved production, and organic amendments. On the one hand, extracting harvested products ($F_{out} > 0$) lowers CSE because a lower fraction of GPP remains in the system. For example, assuming a crop harvest index ranging from 25 to 50% of aboveground biomass (e.g., Unkovich et al. 2010) and a 30% allocation to roots, the percentage of NPP harvested and the corresponding reductions in CSE would range from 17 to 33%. On the other hand, management may improve CSE by increasing the production efficiency of vegetation (Campioli et al. 2015), or involve addition of organic C to fields ($F_{in} > 0$; e.g., manure or biochar). These C amendments increase CSE for given respiration and harvest rates, not only thanks to their direct effect through $F_{in}$, but also thanks to indirect effects when soil amendments promote plant productivity. However, this positive effect lessens as the amended organic C is respired and soil organic C reaches saturation levels (Stewart et al. 2007)."*

As a modeller I have a very selfish request (but I guess it may help others). I appreciated the section 4.5 but I guess that the majority of the modellers using CUE concept are aware of the limitations presented here. Maybe one or two paragraphs with some concrete recommendations will be helpful. In particular, I am wondering if CUE or CSE at organisms or ecosystem levels should be considered as emerging properties of a given system and if yes it might become an interesting approach to evaluate model by comparing the CUE/CSE observed at the system level.

*Yes, we agree that CUE/CSE are emerging properties of a given system and as such they integrate numerous processes and drivers. We also agree that some more specific recommendations may be helpful – to this end we will add the following paragraph to Section 4.5:*

*"In addition to the correct attribution of changes in CUE to processes or environmental conditions, it remains critical to match the definition of CUE used by empiricists with that implemented in models. Specifically, are the same biosynthesis components (e.g., biomass increment vs. exudate export) accounted for in both empirical estimates of CUE and in the model equations? Are abiotic C exchanges at the ecosystem scale both included in empirical estimates of CSE and described by models? As CUE and CSE represent emerging properties of organisms and ecosystems, they are appealing for model testing, but without a consistent definition, comparisons of model outputs and empirical estimates are not meaningful."*

**References**

Campioli, M., S. Vicca, S. Luyssaert, J. Bilcke, E. Ceschia, F. S. Chapin Iii, P. Ciais, M. Fernandez-Martinez, Y. Malhi, M. Obersteiner, D. Olefeldt, D. Papale, S. L. Piao, J. Penuelas, P. F. Sullivan, X. Wang, T. Zenone, and I. A. Janssens. 2015. Biomass production efficiency controlled by management in temperate and boreal ecosystems. Nature Geosci **8**:843-846.

Frey, S. D., V. Gupta, E. T. Elliott, and K. Paustian. 2001. Protozoan grazing affects estimates of carbon utilization efficiency of the soil microbial community. Soil Biology & Biochemistry **33**:1759-1768.

Krausmann, F., K. H. Erb, S. Gingrich, H. Haberl, A. Bondeau, V. Gaube, C. Lauk, C. Plutzar, and T. D. Searchinger. 2013. Global human appropriation of net primary production doubled in the 20th century. Proceedings of the National Academy of Sciences of the United States of America **110**:10324-10329.

Ladd, J. N., L. Jocteurmonrozier, and M. Amato. 1992. Carbon turnover and nitrogen transformations in an alfisol and vertisol amended with $^{14}$C[U]glucose and $^{15}$N ammonium sulfate. Soil Biology & Biochemistry **24**:359-371.

Lenhart, K., B. Weber, W. Elbert, J. Steinkamp, T. Clough, P. Crutzen, U. Poschl, and F. Keppler. 2015. Nitrous oxide and methane emissions from cryptogamic covers. Global Change Biology **21**:3889-3900.

Porada, P., B. Weber, W. Elbert, U. Poschl, and A. Kleidon. 2013. Estimating global carbon uptake by lichens and bryophytes with a process-based model. Biogeosciences **10**:6989-7033.

Stewart, C. E., K. Paustian, R. T. Conant, A. F. Plante, and J. Six. 2007. Soil carbon saturation: concept, evidence and evaluation. Biogeochemistry **86**:19-31.

Unkovich, M., J. Baldock, and M. Forbes. 2010. Variability in harvest index of grain crops and potential significance for carbon accounting: examples from Australian agriculture. Pages 173-219 *in* D. L. Sparks, editor. Advances in Agronomy, Vol 105.

---

## Author Comment (AC3) · 16 Aug 2018

**General Response to Referees' Comments**

Three anonymous referees have provided generally positive comments on our manuscript, with some constructive critiques that we plan to address as explained below (responses are written in *italic*; modified text in quotation marks). The three main comments are as follows:

- Reviewer #1: ambiguous definition of C-use efficiency (CUE) for organisms and communities. The reviewer correctly argues that CUE should be defined as 1-respiration rate/uptake rate. This means that biological products such as exudates should be accounted for in the CUE definition, differently from gross growth or biomass production efficiency in which net biomass increments only are accounted for. We will revise our definition according to the reviewer's suggestion.
- Reviewer #2: lacking discussion on human impacts on ecosystem scale efficiencies. This is an excellent suggestion as it prompted us to generalize the definition of C storage efficiency (CSE). We will include anthropogenic C fluxes in our definition of CSE and comment on the role of ecosystem management as a driver of change for CSE.
- Reviewer #3: lack of clarity in some definitions. We will revise the text to clarify ambiguous definitions, and change Table 3 to indicate which components of the C cycle represent inputs and outputs in each of the systems/scales considered.

Detailed responses are reported in the individual rebuttal letters.

**Anonymous Referee #3**

The manuscript is descriptive without a very extensive data analysis.  However, the synthesis is new (I've never read about such large comparison of CUE across biological systems and biological scales) and interesting (I particularly like the fundamental Fig. 6). So, I think the manuscript is suited for publication without a data re-analysis.

*We agree that we did not present a very extensive data analysis (we will add some clarifications to this regard in Section 3 "Data collection and analysis"; see response to Reviewer 2). Our aim is more modest – we show variability in the data and some patterns that are illustrative of underlying dynamics or drivers and, as also stated above, the main novelty is represented by the synthesis of definitions and the comparison of many conceptually related type of data.*

However, there are key points that need to be improved (do not underestimate them, even are just text improvements). The Theory (paragraph 2) and definitions are fundamental in this paper, yet are not fully clear.

*Thanks for pointing to these issues, which we will address as explained below.*

*for all biological systems, you use the term CUE. However, as well reported in Table 2, for some systems other terms are used. Furthermore, CUE is associated to a specific variable/system (plant and community CUE=NPP/GPP). It would have been much less confusing (and more relevant) if you were proposing an overarching (new) efficiency term, and not 'impose' the one used for some systems to all cases.

*The acronym CUE was first used in plant science in the mid-1990s and later it has been proliferating across disciplines (see Table 2). Therefore, given the widespread use of this acronym, it seems appropriate to compare the different definitions proposed and choose one that represents the conceptual consensuses across these disciplines. For this reason, we would prefer to avoid introducing new terms, but we also acknowledge that 'imposing' a single term may create confusions (as also pointed out by Reviewer 1). We will give a more rigorous definition of CUE as opposed to gross growth efficiency and biomass production efficiency (please see also to this regard our response to Reviewer 1). By defining CUE as the ratio of all biosynthesis products (new biomass and compounds that will be released e.g., as exudates), previous definitions become comparable. In fact, in plant science and in papers on ecosystem C fluxes, CUE is defined as NPP/GPP, consistent with our revised definition. The microbial CUE is also already defined in this way by some (but not all) authors, but given the probably lower role of microbial exudates compared to plant exudates (though evidence is lacking), the discrepancy is mainly semantic. Hence the revised definition does not contradict previous work, but offers an improvement that can be embraced in future publications.*

*To this end, we will revise the paragraph on CUE definition as (see also responses to Reviewer 1):*

*"We now define CUE at the organism level as the ratio between the rate of production of biomass and products (G+EX), and the rate of C uptake (U),*

$$CUE = \frac{G+EX}{U} = 1 - \frac{EG+R}{U}.$$  *(5)*

*As a result, the mass balance equation can be rewritten as,*

$$\frac{dC}{dt} = CUE \times U - EX - T. \tag{6}$$

*With this definition, CUE represents the fraction of C taken up allocated to biosynthesis (biomass and products that will be eventually exuded), but excluding respired and egested C, which do not contribute to biosynthesis. Including exudates such as enzymes and polymeric compounds in the CUE definition is motivated by the clear fitness advantage these products have for the organism…"*

*your attempt of generalization (paragraph 2) is not always easy to follow because each domain (plant, micro-organisms, ecosystems etc.) has his own specific definitions and terminology. It would be easier if you, before generalize (so before paragraph 2.1), describe the specific ways CUE is calculated for each of the five 'scales' you synthesize in Fig 6, thus an extension of Table 2. And then, when you generalize, make several examples. For instance, what is 'Output' (Eq. 1) for the five scales?

*The structure suggested by the Reviewer – extensive discipline-specific definitions before the theory section – is indeed a good alternative to our structure. However, it has a shortcoming: it requires defining many terms that would appear as conceptually different (because they are called in different ways across disciplines). Presenting first a general theory allows avoiding this shortcoming. Extensive definitions and discussions on how C fluxes are measured and interpreted at the various scales is presented in the Supplementary Information. Unfortunately, we cannot include them in the main text also due to space limitations; examples are also provided in Figures 4 and 5.*

*We will implement the suggestion to clarify what 'input' and 'output' represent across scales. Specifically, we will revise Table 3 by restructuring the columns and highlighting which components of the C cycle are inputs or outputs for each system and scale. The table heading will change as follows:*

| System | Inputs | | | Outputs | | | |
|---|---|---|---|---|---|---|---|
| | $U$ | $G$ | $F_{in}$ | $R$ | $T$ | $EX$ (and $EG$) | $F_{out}$ |

*There are the definitions used in the field-specific literature (Table 2) and you add other definitions: CUEapparent, AE, NGE, GGE, CUE ecosystem (extremely confusing: NPP/GPP or NEP/GPP?). Make some choices (can the definitions be reduced?) and clarify.

*AE, NGE, GGE, and ecosystem CUE are all terms currently in use. In particular, the first three acronyms are typically adopted in the literature on animal physiology (Sterner and Elser 2002). The term 'apparent' in association with CUE or equivalent efficiencies is also already in use to indicate when the efficiency is estimated from data, but the obtained values are not 'true' efficiencies due to confounding factors (Hagerty et al. 2014). On purpose, we present all the definitions currently in use and compare them conceptually and mathematically.*

*We are sorry that the way ecosystem CUE – a term already in use as well (Fernandez-Martinez et al. 2014) – is defined in our manuscript was not clear. Ecosystem CUE = NEP/GPP, as reported in Table 2 and Figure 2 in the main text, and Section 1.7 in the Supplementary Materials. Perhaps, part of the confusion arises because the ratio NEP/GPP refers to the biotic component of C exchanges at the ecosystem scale, but we also define a C storage efficiency*

*that includes abiotic (or anthropogenic) C exchanges. The relevant explanations are reported in section 1.7 of the Supplementary Materials:*

*"At the ecosystem level, both CUE of the biotic components and CSE can be defined. When focusing on the biotic components, the only input $U = GPP$ and the only output is respiration (assuming exudates are re-cycled), which comprises autotrophic and heterotrophic terms. Net ecosystem productivity (NEP) is thus defined as the difference between GPP and the total respiration ($R = R_a + R_h$), and ecosystem CUE can be written as,*

$$CUE_{ecosystem} = \frac{Net\ ecosystem\ productivity}{Gross\ primary\ productivity} = \frac{NEP}{GPP} = 1 - \frac{R}{GPP} = 1 - \frac{R_a + R_h}{GPP} =$$

$$CUE_{plant\ community} - \frac{R_h}{GPP}, \tag{1}$$

*where the first equality is used for empirical estimation of ecosystem CUE (Fernandez-Martinez et al. 2014), whereas the last equality links ecosystem CUE to the vegetation CUE (=NPP/GPP) and the heterotrophic respiration to GPP ratio. When including abiotic components and thus lateral abiotic fluxes, Eq. (10) in the main text can be used to obtain,*

$$CSE_{ecosystem} = 1 - \frac{R_a + R_h + F_{out}}{GPP + F_{in}}. \tag{2}$$*"*

*For some cases, you mention the possibility of negative CUE, but for plant (CUE=NPP/GPP) it would not be possible because NPP>0 or =0). Similarly, turnover has a meaning for microbes and another for plants (e.g. in forests, turnover refers to the annual leaves, branch or root turnover and it is added in NPP, Clark et al 2001 Ecological Application 11(2), pp. 356–370).

*We agree that turnover has different meaning in different context, but in all cases it represent a loss of viable biomass due to senescence, damage, or herbivory. Mathematically, it might cause negative 'apparent' CUE when growth is lower than biomass losses. The paper by Clark et al. (2001) describes how NPP can be estimated. Turnover terms are considered in that paper because of the methodology used to measure net biomass increments over a given interval. This methodology does not capture organic material that was fixed but lost during that period. Hence, NPP remains defined as GPP minus autotrophic respiration, and not GPP minus respiration plus turnover. However, empirical estimates of NPP require accounting for C loss via various turnover mechanisms to correctly count the actual biomass increment. Therefore, unless we misinterpreted the comment, we would prefer not to make changes to the manuscript in response to this comment.*

Other remarks
*Your main key syntheses were (from abstract): (i) CUE increases with improving growing conditions, (ii) CUE decrease due to turnover, (iii) CUE decreases with increasing biological and ecological organization. Write them also in Conclusions (instead of generic sentences from L497 to L505) with the key reasons/explanations.

*This is a good suggestion, which we implemented. The Conclusions will be almost completely rewritten to incorporate the main closing arguments on both terminological issues raised by the other reviewers, and trends from our data synthesis:*

*"We have synthesized definitions of and explored variations in the efficiency of C use by organisms, communities and ecosystems, and in the efficiency of C storage in soils and sediments. This synthesis highlighted conceptual similarities in the way these efficiencies are*

*defined across disciplines, and some common terminological and interpretation issues. In particular, the same term CUE (but also other synonyms) is often used at organism-to-community scales to indicate actual C-use efficiency (Eq. (5)), apparent C-use efficiency (related but not equal to CUE, Eq. (8)), and gross growth efficiency (Fig. 1). This mixed use may cause misinterpretations, as it is not clear whether turnover and biological products are included in the CUE calculations. Similarly, at the ecosystem scale the term CUE is used without specifying whether abiotic and anthropogenic fluxes are accounted for. For improved clarity, we suggest to always define how CUE is estimated with particular attention to C exchanges other than biomass increments and respiration.*

*Our synthesis shows that turnover deflates 'apparent' CUE estimates, but not 'actual' CUE calculated as biosynthesis over C uptake ratio. Improving growing conditions generally increases CUE and CSE because it promotes growth processes over C loss processes. Finally, CUE tends to decrease with the level of ecological organization – e.g., from rapidly growing individual organisms to natural communities and ecosystems – as less efficient individuals are considered in communities and more heterotrophic components are sequentially added to the system. Because CUE and CSE are outcomes of a wide spectrum of processes, they are expected to be flexible and to respond to both biological (e.g., trends in growth vs. respiration) and physical controls (e.g., C transport and environmental conditions). As such – and provided that empirical and model definitions of these efficiencies are consistent – they are useful indices of changes in the C cycle through time and space and could be employed to benchmark short- (in the case of CUE) and long-term predictions (CSE) of soil and ecosystem models."*

\*L320-321: as in plants CUE= NPP/GPP and seed production is accounted in NPP, I do not understand your point

*We will re-write this sentence to clarify that crops (not a generic "plant" as stated in the original manuscript) stop photosynthesizing and therefore CUE decreases:*

*"Similarly, crops maintain a high CUE until they stop growing vegetative tissues, which senescence while resources are translocated to seeds"*

\*L503 can be moved above where you discuss applicability of CUE values.

*This sentence will be moved as suggested.*

\*You do not make reference to Campioli et al 2015 Nat Geo. However, that synthesis can be useful, not only for the additional dataset on CUE (that they consider BPE there) but for comparison of ecosystems of different complexity (e.g. natural grassland vs. cropland monoculture). Also there are various suggestions for practical use of CUE/BPE in that paper.

*Thanks for pointing out to this publication, which is now cited and used extensively in our manuscript.*

**References**

Fernandez-Martinez, M., S. Vicca, I. A. Janssens, J. Sardans, S. Luyssaert, M. Campioli, F. S. Chapin Iii, P. Ciais, Y. Malhi, M. Obersteiner, D. Papale, S. L. Piao, M. Reichstein, F. Roda, and J. Penuelas. 2014. Nutrient availability as the key regulator of global forest carbon balance. Nature Clim. Change **4**:471–476.

Hagerty, S. B., K. J. van Groenigen, S. D. Allison, B. A. Hungate, E. Schwartz, G. W. Koch, R. K. Kolka, and P. Dijkstra. 2014. Accelerated microbial turnover but constant growth efficiency with warming in soil. Nature Climate Change **4**:903-906.

Sterner, R. W., and J. J. Elser. 2002. Ecological stoichiometry. The biology of elements from molecules to the biosphere. Princeton University Press, Princeton and Oxford.

---

## Author Response (AR1)

**Response to the Editor's Comments**

Thanks for the opportunity to submit a revised manuscript following the suggestions of the reviewers. We have already detailed how we planned to revise our manuscript in the responses to the Reviewers' comments posted in the public discussion forum. In this letter, we only describe how the planned changes have been implemented, and leave our arguments for those changes in the open discussion to avoid unnecessary repetitions. Differences between planned and actual changes are mostly editorial, but we also mildly modified the data analysis. Thus, the content of this letter is very similar to that of the previously submitted reviewer response letter.

In addition to the specific responses, we streamlined the text (while also being more careful in distinguishing between CUE and GGE or BPE) and improved most figures (notably adding more complete CUE definitions in Figures 1-2, and changing Figures 5f, 6, and 7, where vegetation CUE values are now based on the dataset by Campioli et al. (2015)). Former Figure S1 was removed as we deemed it unnecessary (the schematic figures in the main text already contained the relevant information). We also modified the way data is presented to make datasets more comparable. Leaf CUE is now estimated at the daily time scale, similar to CUE of nonvascular vegetation (by simply assuming an equal duration of day and night), and to estimate ecosystem CUE, we calculated long-term averages of the C flux data from Luyssaert et al. (2009). This resulted in a smaller, but more self-consistent CUE database compared to the original version. Overall, we feel this revised version is more rigorous and better achieves our goal to provide a consistent definition of CUE across disciplines.

Point-by-point responses are written in italic below.

**Anonymous Referee #1**

Manzoni et al reviewed and synthesized patterns in carbon use efficiency (CUE) across scales. This is a large effort that can help reconciling previously identified differences in CUE. The authors go into the details of the different definitions that have been used and clarify some of the misunderstandings in the past. I think this could become an important contribution to the field, as differences in definitions and equations for CUE have been mostly ignored and confusion exists on what CUE should reflect. However, I do not fully agree on the presented definitions and think the manuscript still fails to fully resolve discrepancies. The current manuscript does not accurately represented what CUE is, where the term originates from and how it has been used in the past. As the manuscript reads now, I find it a missed opportunity to resolve the confusion that is associated with this topic.

*We hope that with this revised version, we come closer to providing a synthesis to resolve inconsistencies in the definitions of CUE.*

About the definitions:
From a plant perspective, theory indicates that CUE = NPP/GPP, with NPP = the synthesis of organic compounds = GPP – R. Hence, CUE = 1-R/GPP. (NPP = net primary production; GPP = gross primary production, R = respiration). This corresponds more or less to equation 3 (CUE = 1-outputs/inputs) used by Manzoni et al. However, Manzoni et al consider egestion (EG) and exudation (EX, including symbionts) as part of the outputs, and not as part of NPP. Consequently, the CUE considered here is actually biomass production efficiency (biomass production/inputs = BPE) instead of CUE. Both CUE and BPE are meaningful terms – CUE focusses on the C cycle, while BPE targets the biomass that is produced. In the past, both terms have been rarely distinguished though and they have also not been used consistently.

Correct use can be critical, however, as CUE and BPE may respond differently to environmental changes. For example, an increase in BPE could be associated with un-altered CUE if the partitioning to EX is the sole responsible of the change in BPE (i.e. R unchanged). Such understanding becomes important for example when comparing models with observations. Model evaluation assuming observed BPE = modelled CUE (=1-R/GPP) can lead to serious flaws, as illustrated by the following hypothetical example. Assuming modeled CUE should equal observed BPE, a (hypothetical) decrease of BPE with increasing CO2 concentration would suggest and increase in R/GPP whereas in reality the decrease in BPE may be solely due to an increase of EX while R/GPP, and hence true CUE, remain unaltered. In this hypothetical example, adjusting the model to reflect observed BPE in modeled CUE would lead to an overestimation of the response of CUE and R to elevated CO2.

*The distinction between CUE and BPE (we use the term 'gross growth efficiency' instead) has been clarified throughout the manuscript, as explained in the following.*

The above problem related to the assumption that BPE = CUE is more prominent at some levels (e.g. vegetation) than at others (e.g. bacteria). Hence, differences among levels may in part be due to differences in the definition used. This is somewhat acknowledged by the authors, but it would be much clearer and more accurate if BPE and CUE were clearly distinguished throughout the manuscript and if it was made clear in the figures and tables where BPE is calculated, where CUE is calculated and perhaps also where BPE~CUE.

*Following the reviewer's suggestion, the definition of CUE has been modified as:*

*"We now define CUE at the organism level as the ratio between the rate of production of biomass and products (G+EX), and the rate of C uptake (U),*

$$CUE = \frac{G+EX}{U} = \frac{A-R}{U} = 1 - \frac{EG+R}{U}.$$

*As a result, the mass balance equation **Error! Reference source not found.** can be rewritten as,*

$$\frac{dC}{dt} = CUE \times U - EX - T = GGE \times U - T.$$

*With this definition, CUE represents the fraction of C taken up that is allocated to biosynthesis (biomass and products that eventually be exuded), but excluding respired and egested C, which do not contribute to biosynthesis. Including exudates such as enzymes and polymeric compounds in the CUE definition may be motivated by the clear fitness advantage these products have for the organism. Moreover, C storage compounds and osmolytes are also regarded as 'biomass', as they would be measured as cellular material.*

*Other measures of C conversion efficiency have been proposed (Fig. 1) (Sterner and Elser 2002): i) assimilation efficiency (AE = A/U = assimilation/uptake), ii) net growth efficiency (NGE = G/A = net growth/assimilation), and iii) gross growth efficiency (GGE = G/U = AE × NGE = net growth/uptake, see the last equality on the right hand side of Eq. **Error! Reference source not found.**). The GGE can be regarded as a biomass yield or production efficiency, as it considers respired, egested, and exuded C as lost from the organism (Payne 1970, Manzoni et al. 2012, Campioli et al. 2015), different from CUE, which includes exuded C as a product of the C conversion.*

*The CUE definition in Eq. **Error! Reference source not found.** is consistent with previous work on plant C budgets (Thornley and Cannell 2000), but it differs from definitions often used for soil microorganisms where only biomass synthesis is considered and CUE = GGE (Manzoni et al. 2012, Geyer et al. 2016). It is thus important to emphasize that CUE as defined in Eq. **Error! Reference source not found.** is in general higher than GGE. The difference between GGE and CUE is relevant when EX is large, as in the case of organic C exchanges between roots and plant symbionts (Hobbie 2006, Ekblad et al. 2013), or in anaerobic metabolism (Šantrůčková et al. 2004). In the oceans, 10-30% of microbial production is released as dissolved organic C, but this figure also includes dissolved C from microbial turnover (Benner and Herndl 2011, Jiao et al. 2014). For soil microbial communities, the extent of the extracellular enzyme and polysaccharide synthesis is unknown but presumably small compared to the other rates involved, at least in aerobic soils where CUE≈GGE (Frey et al. 2001, Šantrůčková et al. 2004). Therefore, making the distinction between GGE and CUE is less important in these systems (for further discussions in this context, see Geyer et al. 2016)."*

*Further comments in response to the concerns raised by the Reviewer have been added in the Results and Discussion section:*

*"The effect of increasing exudation rate on CUE varies depending on how such increases are realized. If the increase in EX is fuelled by a correspondingly higher U, CUE also increases; however, if the increase in EX occurs at the expenses of G, such that G+EX is*

*constant for a given U, CUE will not be affected. In both scenarios, higher EX decreases the net biomass production, and hence lowers GGE. For example, consistent with these expectations, the microbial CUE values of an aerobic soil (where exudation was negligible) and an anaerobic soil (where exudation was ≈2/3 of the net biomass increment), were comparable (respectively 0.73 vs. 0.70), because the sum of exudation and biomass production were similar (Šantrůčková et al. 2004). However, the GGE of the aerobic soil was much higher than in the anaerobic soil (0.72 vs. 0.43)."*

*A new table has been added in the Supplementary Materials:*

**Table S3. Comparisons of definitions of C-use efficiencies.**

| Definitions in this work | Context | Alternative definitions in published literature | Source |
|---|---|---|---|
| $CUE_A = 1-O/I$ | Soil microbial communities | Ecosystem-scale efficiency of microbial biomass synthesis and recycling of necromass/exudates ($CUE_E$) | (Eq. 2 in Geyer et al. 2016) |
| $GGE = G/U$ | Animals and microorganisms | Gross growth efficiency (GGE) | (Sterner and Elser 2002) |
| | Microbial communities | Carbon use efficiency (CUE) | (Eq. 2 in Manzoni et al. 2012) |
| | Soil microbial communities | Community-scale efficiency of microbial biomass synthesis ($CUE_C$) | (Eq. 1 in Geyer et al. 2016) |
| | Individual plants | Carbon use efficiency (CUE) | (Gifford 1995) |
| | Plant communities | Biomass production efficiency (BPE) | (Campioli et al. 2015) |
| $CUE = 1-R/U$ | Soil microbial communities | Community-scale efficiency of microbial biomass synthesis when EX≈0 (also denoted as $CUE_C$) | (Figure 3 in Geyer et al. 2016) |
| | Plant communities | Carbon use efficiency (CUE = NPP/GPP) | (Cannell and Thornley 2000) |

Specific comments: l.26 and l.50: I don't think biomass production/C uptake is the consensus definition of CUE (see above). Intro: I suggest to review the history of the definitions for CUE more elaborately. Where was it first used, what was the exact definition, how have definitions been applied in different fields...

*The Introduction was amended as:*

*"For biological systems (organs, individual organisms, or even entire communities), CUE is defined as the ratio between the amount of C allocated to biosynthesis (new biomass and biological products, including e.g., exudates) and the amount of C taken up. While the term CUE was proposed in the mid-1990s in the context of plant C balances (Gifford 1995), other terms – e.g., 'growth yield' – referring to the efficiency of substrate conversion into biomass had been in use since the early 1900 (Monod 1949). Now, efficiency definitions are proliferating across many disciplines in biology, ecology, and Earth sciences. While some of these definitions are comparable (and all are deceptively simple), subtle differences often*

*emerge, partly due to conceptual and methodological advances that allow quantification of previously ignored C exchanges. These differences make interpretation of results difficult and complicate cross-disciplinary comparisons.*

*The main difficulty is to unambiguously define what represents growth, release of extracellular compounds or C storage, and reconcile conceptual definitions with empirical estimates (Clark et al. 2001, Chapin et al. 2006, Geyer et al. 2016)..."*

l. 160: clearly define the difference between uptake and assimilation to help the reader in following the different equations

*A clarification was added: "where U is the uptake rate, U-EG is the assimilation rate (i.e., A in Fig. 1), and G is the net growth rate."*

l.175: define overflow respiration. Ion uptake respiration is not mentioned. Is it considered part of growth respiration? See for example Lambers et al 1983, Physiologia Plantarum, 58: 556-563.

*A clarification was added: "Respiration in Eq. **Error! Reference source not found.** can be further broken down into growth ($R_{growth}$), maintenance ($R_{maintenance}$), and overflow ($R_{overflow}$) components, the latter including futile cycles and compensation of stoichiometric imbalances that are activated when C cannot be used for growth or maintenance (Russell and Cook 1995, Cannell and Thornley 2000, Thornley and Cannell 2000, van Bodegom 2007).".*

l.200: replace 'reduces' with 'can be simplified to'.
l.222: add 'and to EX' after 'exports to other parts of the plant'.
l.295: I suggest to replace 'lower estimates of CUE' by 'an underestimation of CUE'.
l.444: 'for a given uptake rate' seems more logical than 'for a given respiration rate'.

*We have either implemented the suggested changes, or the sentences where the ambiguities were have been heavily modified during re-writing.*

I think the authors missed some relevant publications. Cotrufo et al 2013 (Global Change Biology 19, 988-995) discuss the influence of substrate quality on microbial substrate use efficiency (another alternative for CUE), and consequences for soil C storage. This framework deserves at least a mention.

*A comment was added citing the suggested paper: "It could be argued that with more efficient organisms, the ecosystem-level CUE would increase, resulting in larger C accumulation (for soil systems, see Cotrufo et al. 2013). There is indeed evidence that microbial communities with higher CUE enhance soil C storage in terrestrial systems (Kallenbach et al. 2016).".*

Campioli et al 2015 (Nature Geoscience 8, 843-846) provide an update of Luyssaert et al 2007 and Vicca et al 2012 (both cited in the manuscript), and include also other vegetation types than forests. Data are provided in the supplementary files. I suggest considering including these data, or at least refer to them.

*We now refer to the paper by Campioli et al. (2015) and use their dataset in Figures 5, 6, and 7.*

Table 2: Cannell and Thornley 2000 actually used the definition CUE = 1- Ra/GPP.

*We have moved the citation to Cannell and Thornley (2000) to the correct position in the table, next to DeLucia et al. (2007), and cited here Gifford (1995) instead.*

DeLucia et al 2007 used data on biomass production/GPP but termed it NPP/GPP (hence ignoring other NPP components such as exudates and symbionts). This is part of the confusion and I suggest the authors take the opportunity to clarify this.

*The Supplementary information was amended as:*

*"As shown in Eq. (5) in the main text, plant community CUE should be calculated by including both net biomass increments and exudation rates. When only net biomass increments are available, the terms gross growth efficiency (GGE) or biomass production efficiency (BPE) are more accurate (as in Vicca et al. 2012, Campioli et al. 2015). BPE estimates are reported in an extensive global database for forest sites, including direct measurements, indirect estimates (derived from measurements of other C fluxes) and model results (Luyssaert et al. 2007). This dataset has been recently expanded to grasslands and croplands (Campioli et al. 2015) (data used in Fig. 5-7)."*

Fig. 2: CUEplant is defined as NPP/GPP, but NPP is undefined. In line with my earlier comments, I suggest to clearly define NPP.

*We have now clearly defined NPP in the main text and in the SI.*

Figs. 6 and 7: clarify where the data originate from (refer to SI)

*We now refer to the SI by adding: "Data sources are described in the Supplementary Information."*

**Anonymous Referee #2**

The manuscript submitted by Manzoni et al. is a review associated to a database analyse around the concept of carbon use efficiency and carbon storage efficiency.
The quality of the manuscript is very high and I particularly appreciated the effort of the authors to gather data from very different sources to have a broad view of the CUE/CSE concept. The writing is excellent and despite the complexity of the question the authors succeed to make a clear and easy to read document. I am convinced that this paper will be provide an important contribution to the literature and since it deals with data coming from plants, soil, ocean, etc. at different spatiotemporal scales it is of broad interest.
I may have few minor comments to try to make the manuscript even more attractive.

Section 3. Can you provide a bit more details on the methods used to collect the data (e.g. keywords used in ISIWEB).

*We have added a paragraph in Section 3 "Data collection and analysis":*

*"To compile the new data collections, we conducted an online search using ISI Web of Science and Google Scholar with keywords including various synonyms of CUE or CSE. We also gathered publications following relevant references in articles and books, aided by the expert knowledge of the authors. Due to the enormous variability in terminology used across disciplines, and the fact that in many cases CUE or CSE were not reported in the papers (but only C exchange rates to calculate them), a systematic search was not feasible. Nevertheless, while not exhaustive, our selection of publications covers a broad range of conditions for each subset of data, enabling detection of general patterns across disciplines and scales."*

Section 4.1 You cite two studies as example but some methodological details are missing to fully understand your arguments (what kind of carbon added (litter, glucose...) or how long was the incubation for instance).

*Details have been added in the main text to indicate the added compound used (glucose in both cases), and further clarifications have been added in the caption of Figure 4:*

*"Lower turnover rates were caused by lower mortality in the first 3 days of incubation compared to the day 112 (Ladd et al. 1992), or by lower grazing in the first two days of incubation compared to days 7-8 (Frey et al. 2001). Error bars indicate standard errors of the mean (variability is across three soil types in Ladd et al. (1992) and across replicates and soil types in Frey et al. (2001))."*

*and in the caption of Figure 5:*

*"The central panels show decreasing CUE when (c) the C substrate is consumed (moving right to left along the abscissa) during 12 (glucose) and 71 (cellulose) day incubations (Öquist et al. 2017) or (d) resource availability (as the ratio of salicylic acid C to biomass C) is low (Collado et al. 2014)."*

I missed some words on the anthropogenic effect on ecosystems CSE. In all the manuscript you compared different types of ecosystems but it may be interesting to compare systems

highly managed like cropland or European forest and grassland with a substantial fraction of the NPP appropriated by humans (see Krausmann et al. (2013) for instance).

*This topic is now discussed in Section 4:*

*" A large fraction of land and of marine systems is managed to extract food and fibre to support a growing human population (Krausmann et al. 2013). Management of ecosystems has two contrasting effects on CSE, depending on the balance of harvest removal, improved production, and organic amendments. On the one hand, extracting harvested products ($F_{out} > 0$) lowers CSE because a lower fraction of GPP remains in the system. For example, assuming a crop harvest index ranging from 25 to 50% of aboveground biomass (e.g., Unkovich et al. 2010) and a 30% allocation to roots, the percentage of NPP harvested and the corresponding reductions in CSE would range from 17 to 33% (Eq. **Error! Reference source not found.**). On the other hand, management may improve CSE by increasing the production efficiency of vegetation (Campioli et al. 2015), or involve addition of organic C to fields ($F_{in} > 0$; e.g., manure or biochar). These C amendments increase CSE for given respiration and harvest rates, not only thanks to their direct effect through $F_{in}$, but also thanks to indirect effects when soil amendments promote plant productivity. However, this positive effect lessens as the amended organic C is respired and soil organic C reaches saturation levels (Stewart et al. 2007). "*

As a modeller I have a very selfish request (but I guess it may help others). I appreciated the section 4.5 but I guess that the majority of the modellers using CUE concept are aware of the limitations presented here. Maybe one or two paragraphs with some concrete recommendations will be helpful. In particular, I am wondering if CUE or CSE at organisms or ecosystem levels should be considered as emerging properties of a given system and if yes it might become an interesting approach to evaluate model by comparing the CUE/CSE observed at the system level.

*We have added the following paragraph to Section 4.5:*

*"In addition to the correct attribution of changes in CUE to processes or environmental conditions, it remains critical to match the definition of CUE used by empiricists with that implemented in models. Specifically, are the same biosynthesis components (e.g., biomass increment vs. exudate export) accounted for in both empirical efficiency estimates and in the model equations? Are abiotic C exchanges at the ecosystem scale both included in empirical estimates of CSE and described by models? As CUE and CSE represent emerging properties of organisms and ecosystems, they are appealing for model testing, but without a consistent definition, comparisons of model outputs and empirical estimates are not meaningful. "*

**Anonymous Referee #3**

The manuscript is descriptive without a very extensive data analysis. However, the synthesis is new (I've never read about such large comparison of CUE across biological systems and biological scales) and interesting (I particularly like the fundamental Fig. 6). So, I think the manuscript is suited for publication without a data re-analysis.

*Even if not explicitly requested, we modified the data analysis to improve comparability among datasets in our database, as explained in Section 3:*

*"To facilitate comparisons across datasets, instantaneous CUE values estimated for leaves and non-vascular plant communities were converted to daily values by assuming an equal duration of day- and night-time, and that respiration rates were the same throughout the whole day. Moreover, plant community and ecosystem C fluxes (Luyssaert et al. 2007, Campioli et al. 2015) were averaged first when estimates from different approaches were reported for a given site and year, and second across years to provide long-term mean fluxes."*

However, there are key points that need to be improved (do not underestimate them, even are just text improvements). The Theory (paragraph 2) and definitions are fundamental in this paper, yet are not fully clear.

*for all biological systems, you use the term CUE. However, as well reported in Table 2, for some systems other terms are used. Furthermore, CUE is associated to a specific variable/system (plant and community CUE=NPP/GPP). It would have been much less confusing (and more relevant) if you were proposing an overarching (new) efficiency term, and not 'impose' the one used for some systems to all cases.

*We have revised the paragraph on the definition of CUE definition as (see also responses to Reviewer 1):*

*"We now define CUE at the organism level as the ratio between the rate of production of biomass and products (G+EX), and the rate of C uptake (U),*

$$CUE = \frac{G+EX}{U} = \frac{A-R}{U} = 1 - \frac{EG+R}{U}.$$

*As a result, the mass balance equation **Error! Reference source not found.** can be rewritten as,*

$$\frac{dC}{dt} = CUE \times U - EX - T = GGE \times U - T.$$

*With this definition, CUE represents the fraction of C taken up that is allocated to biosynthesis (biomass and products that eventually be exuded), but excluding respired and egested C, which do not contribute to biosynthesis. Including exudates such as enzymes and polymeric compounds in the CUE definition may be motivated by the clear fitness advantage these products have for the organism. Moreover, C storage compounds and osmolytes are also regarded as 'biomass', as they would be measured as cellular material."*

*your attempt of generalization (paragraph 2) is not always easy to follow because each domain (plant, micro-organisms, ecosystems etc.) has his own specific definitions and

terminology. It would be easier if you, before generalize (so before paragraph 2.1), describe the specific ways CUE is calculated for each of the five 'scales' you synthesize in Fig 6, thus an extension of Table 2. And then, when you generalize, make several examples. For instance, what is 'Output' (Eq. 1) for the five scales?

*We implemented the suggestion to clarify what 'input' and 'output' represent for the various systems we considered. Specifically, we revised Table 3 by restructuring the columns and highlighting which components of the C cycle are inputs or outputs for each system and scale. The table heading has been changed as follows:*

| System | Inputs | | | Outputs | | | |
|---|---|---|---|---|---|---|---|
| | $U$ | $G$ | $F_{in}$ | $R$ | $T$ | $EX$ (and $EG$) | $F_{out}$ |

*There are the definitions used in the field-specific literature (Table 2) and you add other definitions: CUEapparent, AE, NGE, GGE, CUE ecosystem (extremely confusing: NPP/GPP or NEP/GPP?). Make some choices (can the definitions be reduced?) and clarify.

*We have not made specific changes in response to this comment (see our rationale in the response to reviewers' comments in the public discussion).*

*For some cases, you mention the possibility of negative CUE, but for plant (CUE=NPP/GPP) it would not be possible because NPP>0 or =0). Similarly, turnover has a meaning for microbes and another for plants (e.g. in forests, turnover refers to the annual leaves, branch or root turnover and it is added in NPP, Clark et al 2001 Ecological Application 11(2), pp. 356–370).

*We have not made specific changes in response to this comment (see our rationale in the response to reviewers' comments in the public discussion).*

Other remarks
*Your main key syntheses were (from abstract): (i) CUE increases with improving growing conditions, (ii) CUE decrease due to turnover, (iii) CUE decreases with increasing biological and ecological organization. Write them also in Conclusions (instead of generic sentences from L497 to L505) with the key reasons/explanations.

*The Conclusions have been rewritten as:*

*"We have synthesized definitions of and explored variations in the efficiency of C use by organisms, communities and ecosystems, and in the efficiency of C storage in soils and sediments. This synthesis highlighted conceptual similarities in the way these efficiencies are defined across disciplines, and some common terminological and interpretation issues. In particular, the same term CUE (but also other synonyms) is often used at organism-to-community scales to indicate actual C-use efficiency (Eq.* **Error! Reference source not found.***), apparent C-use efficiency (related but not equal to CUE, Eq.* **Error! Reference source not found.***), and gross growth efficiency. This mixed use may cause misinterpretations, as it is not clear whether turnover and biological products are included in the CUE calculations. Similarly, at the ecosystem scale the term CUE is used without specifying whether abiotic and anthropogenic fluxes are accounted for. For improved clarity, we suggest to always define how*

*CUE is estimated with particular attention to C exchanges other than biomass increments and respiration.*

*Our synthesis shows that turnover deflates 'apparent' CUE estimates, but not 'actual' CUE calculated as biosynthesis over C uptake ratio. Improving growing conditions generally increases CUE and CSE because it promotes growth processes over C loss processes. Finally, CUE tends to decrease with the level of ecological organization – e.g., from rapidly growing individual organisms to natural communities and ecosystems – as less efficient individuals are considered in communities and more heterotrophic components are sequentially added to the system. Because CUE and CSE are outcomes of a wide spectrum of processes, they are expected to be flexible and to respond to both biological (e.g., trends in growth vs. respiration) and physical controls (e.g., C transport and environmental conditions). As such – and provided that empirical and model definitions of these efficiencies are consistent – they are useful indices of changes in the C cycle through time and space and could be employed to benchmark short-(in the case of CUE) and long-term predictions (CSE) of soil and ecosystem models."*

*L320-321: as in plants CUE= NPP/GPP and seed production is accounted in NPP, I do not understand your point

*We added a clarification: "Similarly, crops maintain a high CUE until they stop growing vegetative tissues, which senescence while resources are translocated to seeds"*

*L503 can be moved above where you discuss applicability of CUE values.

*This sentence has been moved as suggested.*

*You do not make reference to Campioli et al 2015 Nat Geo. However, that synthesis can be useful, not only for the additional dataset on CUE (that they consider BPE there) but for comparison of ecosystems of different complexity (e.g. natural grassland vs. cropland monoculture). Also there are various suggestions for practical use of CUE/BPE in that paper.

*Thanks for pointing out to this publication, which is now cited and used extensively in our manuscript.*

Table S3. Comparisons of definitions of biological C-use efficiencies for plants and soil microorganisms.

| Definitions in this work | Context | Alternative definitions in published literature | Source |
|---|---|---|---|
| $CUE_A = 1-O/I$ | Soil microbial communities | Ecosystem-scale efficiency of microbial biomass synthesis and recycling of necromass/exudates ($CUE_E$) | (Eq. 2 in Geyer et al. 2016) |
| $GGE = G/U$ | Animals and microorganisms | Gross growth efficiency (GGE) | (Sterner and Elser 2002) |
| | Microbial communities | Carbon use efficiency (CUE) | (Eq. 2 in Manzoni et al. 2012) |
| | Soil microbial communities | Community-scale efficiency of microbial biomass synthesis ($CUE_C$) | (Eq. 1 in Geyer et al. 2016) |
| | Individual plants | Carbon use efficiency (CUE) | (Gifford 1995) |
| | Plant communities | Biomass production efficiency (BPE) | (Campioli et al. 2015) |
| $CUE = 1-R/U$ | Soil microbial communities | Community-scale efficiency of microbial biomass synthesis when $EX \approx 0$ (also denoted as $CUE_C$) | (Figure 3 in Geyer et al. 2016) |
| | Plant communities | Carbon use efficiency (CUE = NPP/GPP) | (Cannell and Thornley 2000) |

695

[Figure]

**Figure S1. Comparison of the efficiencies of C export (exported C/primary production) among terrestrial and aquatic ecosystems. (a) Relation between C export rate and net primary productivity; (b) box plot of C-export efficiencies across ecosystem types. Data for terrestrial vegetation and algal beds/macrophytes is from Cebrian and Lartigue (2004); data for oceanic phytoplankton is from Dunne et al. (2005).**

700